# VEGF amplifies transcription through ETS1 acetylation to enable angiogenesis

Jiahuan Chen[1], Yi Fu[1], Daniel S. Day[2,3], Ye Sun [4], Shiyan Wang[1], Xiaodong Liang[1], Fei Gu[5], Fang Zhang[1], Sean M. Stevens[5], Pingzhu Zhou[5], Kai Li[5], Yan Zhang[6,7], Ruei-zeng Lin[7], Lois E.H. Smith[4], Jin Zhang[5], Kun Sun[8], Juan M. Melero-Martin[7,9,10], Zeguang Han[1], Peter J. Park [2], Bing Zhang[1,8] & William T. Pu[5,10]

Release of promoter-proximally paused RNA polymerase II (RNAPII) is a recently recognized transcriptional regulatory checkpoint. The biological roles of RNAPII pause release and the mechanisms by which extracellular signals control it are incompletely understood. Here we show that *VEGF* stimulates RNAPII pause release by stimulating acetylation of *ETS1*, a master endothelial cell transcriptional regulator. In endothelial cells, *ETS1* binds transcribed gene promoters and stimulates their expression by broadly increasing RNAPII pause release. [34]*VEGF* enhances *ETS1* chromatin occupancy and increases *ETS1* acetylation, enhancing its binding to *BRD4*, which recruits the pause release machinery and increases RNAPII pause release. Endothelial cell angiogenic responses in vitro and in vivo require *ETS1*-mediated transduction of *VEGF* signaling to release paused RNAPII. Our results define an angiogenic pathway in which *VEGF* enhances *ETS1*–*BRD4* interaction to broadly promote RNAPII pause release and drive angiogenesis.

[1] Key Laboratory of Systems Biomedicine, Shanghai Center for Systems Biomedicine, Shanghai Jiao Tong University, Shanghai 200240, China. [2] Department for Biomedical Informatics, Harvard Medical School, Boston, MA 02115, USA. [3] Harvard/MIT Division of Health Sciences and Technology, Cambridge, MA 02139, USA. [4] Department of Ophthalmology, Harvard Medical School/Children's Hospital Boston, Boston, MA 02115, USA. [5] Department of Cardiology, Boston Children's Hospital, Boston, MA 02115, USA. [6] Renji-Med Clinical Stem Cell Research Center, Renji Hospital, School of Biomedical Engineering, Shanghai Jiao Tong University, Shanghai 200127, China. [7] Department of Cardiac Surgery, Children's Hospital Boston, Boston, MA 02115, USA. [8] Department of Pediatric Cardiology, Xin Hua Hospital Affiliated to Shanghai Jiao Tong University School of Medicine, Shanghai 200092, China. [9] Department of Surgery, Harvard Medical School, Boston, MA 02115, USA. [10] Harvard Stem Cell Institute, Harvard University, Cambridge, MA 02138, USA. Jiahuan Chen, Yi Fu and Daniel S. Day contributed equally to this work. Correspondence and requests for materials should be addressed to B.Z. (email: bingzhang@sjtu.edu.cn) or to W.T.P. (email: wpu@pulab.org)

Angiogenesis, the growth of blood vessels from existing vasculature, is central to mammalian organ development and disease. Nearly all steps of angiogenesis are controlled by vascular endothelial growth factor (VEGF), which signals through VEGF receptor 2 (VEGFR2) to activate, among others, the MEK–ERK kinase cascade. Ultimately, VEGF stimulation alters endothelial cell (EC) gene transcription to enable vessel formation[1–3]. However, the mechanisms by which VEGF influences gene expression remains poorly understood.

V-Ets Avian Erythroblastosis Virus E26 Oncogene Homolog 1 (ETS1), the founding member of the E26 transformation-specific sequence (ETS) transcription factor family, is a master regulator of EC gene transcription. Upregulation of ETS1 in quiescent ECs was sufficient to convert them to an angiogenic state, and depletion of *ETS1* impaired vascular development during embryogenesis[4, 5]. The ETS motif is found in nearly all angiogenic transcriptional enhancers, and we previously found that ETS1 and the co-activator p300 co-localize at EC enhancers[3, 6]. However, the mechanism by which ETS1 controls EC gene expression and its potential role in angiogenic signal transduction and downstream transcription remain undetermined.

The transition of RNAPII from a promoter-proximally stalled state to active elongation has recently been identified as a key checkpoint for the transcription of many genes[7, 8]. RNAPII pause release requires Positive Transcription Elongation Factor-b (P-TEFb), a kinase which phosphorylates pausing factors and RNAPII on serine 2 of its C-terminal domain. Bromodomain-containing protein 4 (*BRD4*) and certain transcription factors, such as MYC and NFKB, recruit P-TEFb to transcriptional start sites (TSSs) to release paused RNAPII[9–12]. Although many environmental stress-responsive and developmental genes have promoter-proximally paused RNAPII[13–15], the molecular mechanisms that link environmental stimuli to altered RNAPII pausing at these genes are incompletely understood.

In this study, we identified *ETS1* as a new RNAPII pausing regulator that globally promotes pause release in ECs. This role of *ETS1* to stimulate RNAPII pause release was regulated by VEGF and essential for VEGF angiogenic activity. Together, our study implicates VEGF-stimulated RNAPII pause release as an important regulatory step in angiogenesis. More broadly, our study provides a new and possibly widely applicable mechanistic model by which extracellular stimuli influences RNAPII pausing and gene transcription.

## Results

**Promoter-proximal-ETS1 positively correlated with transcription.** ETS1 is a master transcription factor in ECs and activates angiogenesis[3–5]. To unveil the transcriptional mechanisms by which ETS1 regulates anigogenesis, we analyzed its chromatin occupancy in human umbilical vein endothelial cells (HUVECs) by ChIP-seq before and after *VEGF* stimulation (Fig. 1a, Supplementary Table 1)[3]. The ETS motif was the most significantly enriched motif in these regions, consistent with high quality of these data (Supplementary Fig. 1a). ETS1 was highly enriched at promoters, with 20–28% of bound regions located within 1 kb of TSSs (Fig. 1b, Supplementary Fig. 1b). To determine the relationship of ETS1 to other features of the chromatin landscape, we performed ChIP-seq for histones with post-translational modifications associated with active or repressed transcription, as well as RNAPII. At promoters, ETS1 co-localized with H3K27ac, H3K4me2, H3K4me3 and RNAPII, chromatin features positively correlated with promoter activity[16, 17], but poorly overlapped with H3K27me3, a feature negatively correlated with promoter activity (Fig. 1c, Supplementary Fig. 1c, and Supplementary Table 1). We also found that ETS1 overlapped at promoters with

MYC (Fig. 1c, Supplementary Fig. 1d), recently shown to widely bind promoters to stimulate RNAPII pause release[9–11], and that ETS1 and MYC promoter signals were well correlated (Supplementary Fig. 1e). Using RNA sequencing (RNA-seq) data from the same time course (ref. 3, Supplementary Table 1), we compared ETS1 promoter occupancy to gene transcriptional activity. This analysis revealed that ETS1 preferentially occupied promoters of expressed genes, and infrequently occupied promoters of non-transcribed genes (Fig. 1d).

ETS1 occupancy of the promoters of most expressed genes led us to hypothesize that it positively stimulates gene transcription

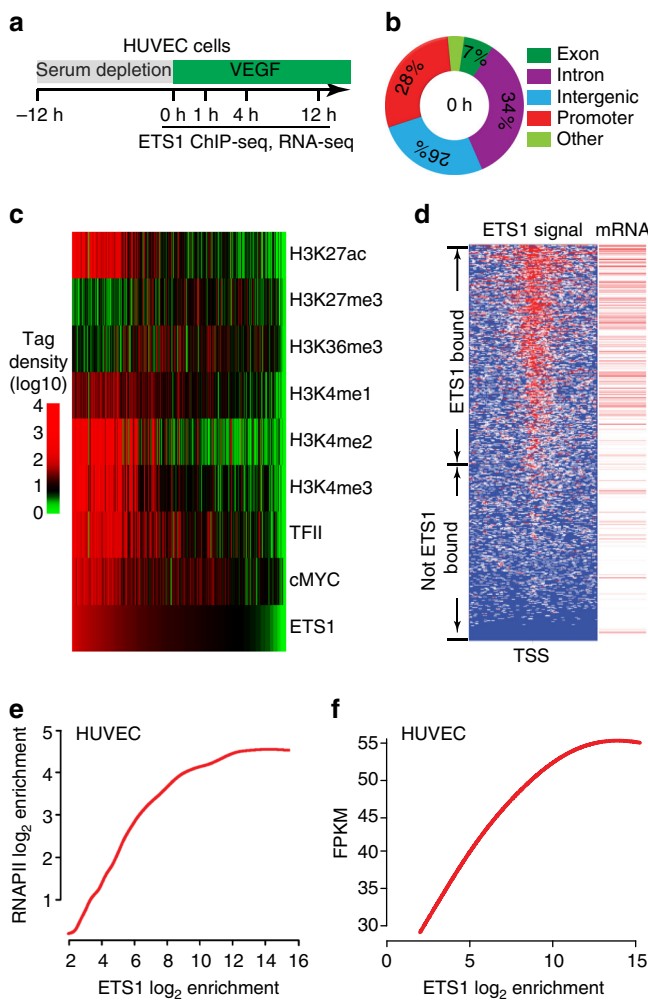

**Fig. 1** ETS1 promoter occupancy and gene expression. ETS1 occupied promoters of most expressed genes, and its promoter occupancy correlated with gene expression. **a** Overview of the experimental design used for in vitro studies. Samples were collected prior to stimulation (0 h) and at 1, 4, and 12 h of VEGF stimulation. **b** ETS1 chromatin occupancy at 0 h with respect to genome annotations. **c** Heatmap of indicated chromatin features at promoter regions at the 0 h time point. Regions are ordered by ETS1 binding strength at 0 h after VEGF stimulation. Features positively correlated with gene expression correlated with ETS1 binding strength. **d** ETS1 signal at TSS region and associated gene expression at the 0 h time point. ETS1 bound most expressed genes. *Left panel* (ETS1 signal): tag heatmap with high ChiP-seq signal shown in *red*. *Right panel* (mRNA): *red lines* indicate expressed genes, as determined by RNA-seq. **e** Correlation plot of promoter ETS1 and RNAPII occupancy at the 0 h time point. **f** Correlation plot of promoter ETS1 occupancy and RNA-seq gene expression at the 0 h time point

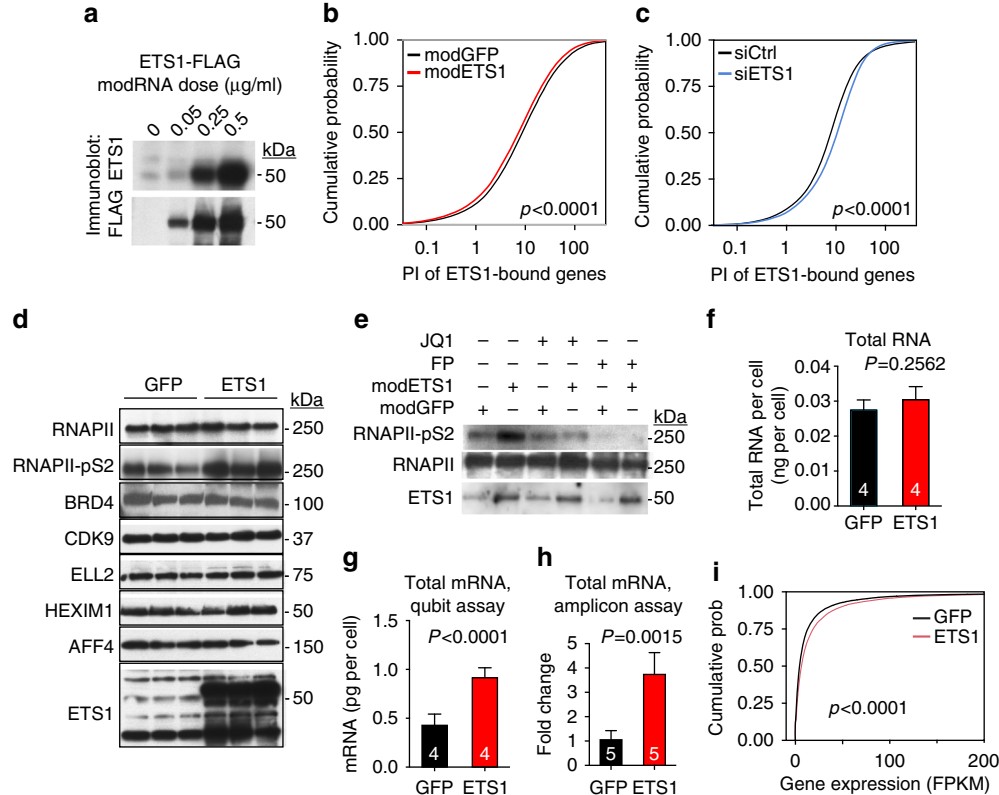

**Fig. 2** *ETS1* stimulated RNAPII pause release. **a** Immunoblot of ETS1 expression in HUVEC cells 12 h after transfection by the indicated dose of *ETS1* modRNA. **b** ETS1 overexpression reduced RNAPII pausing at ETS1-bound promoters. HUVEC cells were treated with *ETS1* or *GFP* modRNA. Pausing index (PI) of ETS1-bound genes, a measure of a gene's RNAPII paused at its promoter, was calculated from RNAPII ChIP-seq performed 12 h after transfection. ETS1 shifted the distribution of ETS1-bound genes to lower PI in treatment compared to control. **c** ETS1 knockdown increased RNAPII pausing at ETS1-bound promoters. Experiment as in **b**, except that cells were treated with control or *ETS1* siRNAs. **d** ETS1 overexpression using modRNA increased actively elongating RNAPII (RNAPII-pS2) but not total RNAPII. HUVEC cells treated with *GFP* or *ETS1* modRNA were analyzed by immunoblot at 12 h. **e** ETS1 overexpression increased actively elongating RNAPII through BRD4 and P-TEFb. *ETS1* modRNA-induced increase of RNAPII-pS2 was blocked by BRD4 inhibitor JQ1 or P-TEFb inhibitor flavopiridole (FP). **f–i** ETS1 overexpression broadly increased mRNA expression. Total RNA **f** or mRNA **g** content per cell were measured by Qubit assay. Alternatively, mRNA was converted to RNA-seq libraries, using external spike-in RNA for normalization to cell number. Relative RNA-seq library yield per cell was measured by quantitative RTPCR. Cumulative distribution plot of RNA abundance per cell. Cumulative distribution plot of gene expression **i** showed that that *ETS1* modRNA broadly increased gene expression. *P*-values were calculated by Student's *t*-test (**f–h**) or by Kolmogorov–Smirnov test **b**, **c**, **i**. *Bar graphs* show mean ± s.d

genome-wide. Consistent with this hypothesis, ETS1 promoter occupancy positively correlated with RNAPII level at the promoter (Fig. 1e) and with gene expression (Fig. 1f). We observed a similar correlation between gene expression and ETS1 promoter occupancy in two additional cell lines (K562 and GM12898 cells; Supplementary Fig. 1f), suggesting generalizability across cell types.

Collectively, our data show that ETS1 is highly enriched at the promoters of expressed genes and is broadly correlated with gene expression in multiple cellular contexts.

**ETS1 promotes RNAPII pausing-release.** The global association between ETS1 and gene expression led us to hypothesize that ETS1 amplifies transcription in ECs by increasing RNAPII pause release. To assess the role of ETS1 in this process, we upregulated ETS1 in HUVECs using modified RNA (modRNA), a highly efficient and non-toxic technique to rapidly express gene products in difficult-to-transfect cells including HUVEC[18, 19] (Supplementary Fig. 2a, b). *ETS1* modRNA dramatically increased ETS1 expression within 12 h, without adversely affecting cell morphology (Fig. 2a). To measure RNAPII pausing genome wide, we used RNAPII ChIP-seq to determine each gene's Pausing Index (PI), the ratio of length-normalized RNAPII signal near its

promoter (−50 bp to +300 bp around TSS) to RNAPII in its gene body (+300 bp to 3 kb past TES, see Methods for details)[11, 13]. Genes with higher PI have a greater fraction of RNAPII paused near their promoter. Compared to control (GFP modRNA), *ETS1* modRNA globally and significantly reduced PI of genes with ETS1 promoter occupancy ($p < 0.0001$, Kolmogorov–Smirnov test; Fig. 2b). Depletion of *ETS1* by siRNA (Supplementary Fig. 2c, d) had the opposite effect of globally and significantly increasing PI at ETS1-occupied genes ($p < 0.0001$, Kolmogorov–Smirnov test; Fig. 2c). Moreover, *ETS1* modRNA did not significantly change the chromatin occupancy of other ETS1 family members (ERG and FLI1) on most loci (Supplementary Fig. 2e) analyzed by ChIP-qPCR. Together, these results demonstrate that ETS1 promotes RNAPII pause release.

Release of paused RNAPII into its actively elongating form is associated with phosphorylation on serine 2 of its C-terminal domain (RNAPII-pS2) by CDK9, a subunit of the essential pause release complex P-TEFb or by BRD4, a protein that recruits P-TEFb and that was recently reported to directly phosphorylate RNAPII-S2[1, 20]. ETS1 overexpression markedly increased RNAPII-pS2, without affecting overall levels of RNAPII or the key pause release regulators CDK9, HEXIM1, ELL2, BRD4, or AFF4 (Fig. 2d). The ETS1-induced increase of RNAPII-pS2 was blocked by flavopiridol (FP), a CDK9 kinase inhibitor, confirming

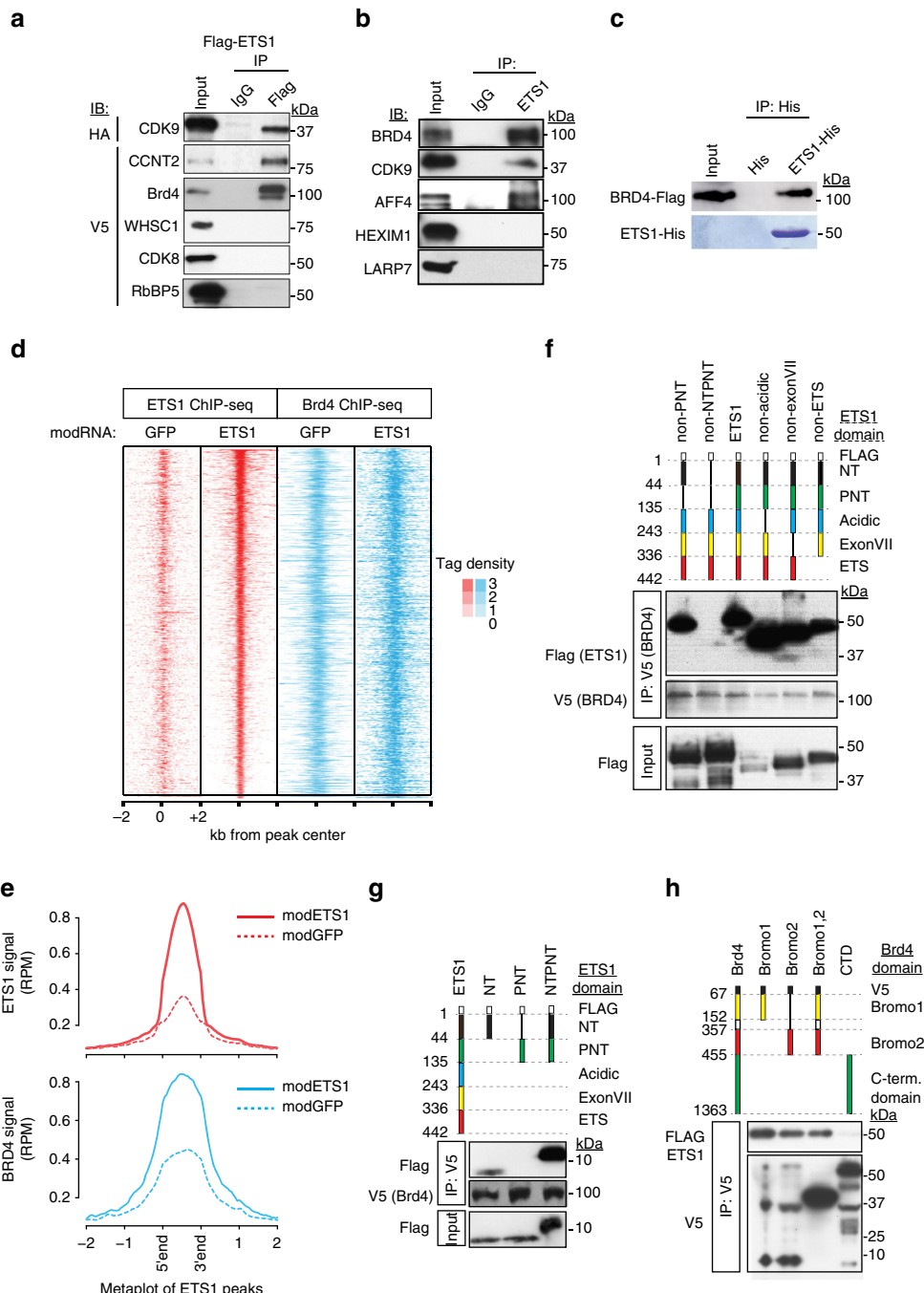

**Fig. 3** ETS1 recruited P-TEFb to chromatin by direct interaction with BRD4. **a** ETS1 co-precipitated P-TEFb and BRD4. 293T cells were co-transfected with indicated expression constructs. ETS1 was immunoprecipitated (FLAG), and interacting proteins were detected with HA or V5 antibodies. **b** ETS1 interacted with active P-TEFb in HUVEC cells. Endogenous ETS1 was immunoprecipitated, and endogenous interacting proteins were detected using specific antibodies. **c** Bacterially expressed, affinity purified ETS1-His bound in vitro transcribed and translated BRD4. **d** ETS1 and BRD4 promoter co-occupancy. HUVEC cells were transfected with *ETS1* or *GFP* modRNA, and ETS1 and BRD4 chromatin occupancy was measured by ChIP-seq. The tag heatmap displays ETS1 and BRD4 signals from ETS1 regions within promoters. ETS1 and BRD4 co-occupied promoters, and increased ETS1 occupancy correlated with increased BRD4 occupancy. **e** Aggregation plots of ETS1 and BRD4 signals shown in **d**. **f, g** *BRD4* binds the ETS1 NT domain. FLAG-tagged ETS1 expression constructs containing the indicated domains were co-transfected into 293T cells with V5-tagged BRD4. ETS1 deletion mutants were detected in BRD4 (V5) immunoprecipitates by FLAG immunoblotting. The NT domain was required for BRD4 binding. **h** BRD4 Bromo domains bind ETS1. V5-tagged expression constructs containing the indicated BRD4 domains were co-transfected into 293T cells with FLAG-tagged ETS1. BRD4 (V5) immunoprecipitates were probed for interacting ETS1 protein by FLAG immunoblotting. Either bromodomain of BRD4 was sufficient for ETS1 interaction

that ETS1 overexpression increases RNAPII-pS2 through this complex (Fig. 2e). JQ1, a small molecule inhibitor of BRD4, also inhibited increased RNAPII-pS2 (Fig. 2e). These data indicate that ETS1 stimulates RNAPII pause release through BRD4 and the active form of P-TEFb.

**ETS1 amplified global transcription of mRNA.** Genome-wide increase of RNAPII pause release is able to increase RNA content globally[10, 11]. To determine if ETS1 had similar activity in ECs, we measured RNA abundance per cell. Total RNA and rRNA content per cell were not significantly different between *ETS1* and control modRNA transfection (Fig. 2f, Supplementary Fig. 2e–g), unlike MYC's observed effect in stem and tumor cell lines[10, 11]. However, mRNA content per cell was significantly increased by *ETS1* modRNA transfection (Fig. 2g). To further evaluate this effect, we prepared RNA-seq libraries from HUVEC cells treated with *ETS1* or *GFP* modRNA. External reference RNA was spiked into isolated total RNA to permit downstream normalization for cell number. Libraries prepared from *ETS1*-treated cells gave reproducibly higher yield after external reference RNA normalization than control cells (Fig. 2h), consistent with overall elevated mRNA per cell. We sequenced these libraries and determined gene expression levels per cell by normalizing to the external spike-in reference mRNA. *ETS1* modRNA broadly increased the expression of most transcripts, with 84% of transcripts having higher expression with ETS1 overexpression (Fig. 2i and Supplementary Fig. 2h; $p < 0.0001$, Kolmogorov–Smirnov test).

Collectively, these data demonstrate that ETS1 globally increases mRNA transcription in ECs through P-TEFb and BRD4-mediated RNAPII pause release.

**ETS1 recruits P-TEFb to promoters by directly interacting with BRD4.** RNAPII pause release is controlled by P-TEFb (CDK9, CCNT1/2) and its positive (BRD4, AFF4), and negative (HEXIM1, LARP7) regulatory factors[8, 21]. To explore mechanisms by which ETS1 stimulates RNAPII pause release, we scrutinized the interaction between ETS1 and these factors. In 293T cells, Flag-ETS1 co-precipitated CDK9, CCNT2, and BRD4 (Fig. 3a). Negative controls WHSC1, CDK8, or RbBP5, unrelated proteins linked to RNAPII initiation or elongation, did not co-precipitate (Fig. 3a), demonstrating the specificity of the co-IP assay. In HUVEC cells as well, ETS1 co-immunoprecipitated P-TEFb component CDK9 and its positive regulatory factors BRD4 and AFF4 (Fig. 3b), demonstrating that these interactions occur in ECs at endogenous protein expression levels. Reciprocal co-IP experiments in HUVECs further confirmed ETS1 interaction with BRD4 and CDK9 (Supplementary Fig. 3a). In contrast, P-TEFb negative regulators HEXIM1 and LARP7 were not co-precipitated (Fig. 3b), indicating that ETS1 interacts with active and not inactive P-TEFb.

To test the hypothesis that ETS1 directly binds with BRD4, we analyzed the interaction between bacterially expressed, His-tagged ETS1 and in vitro translated BRD4. We found that these proteins robustly interact in vitro (Fig. 3c). To explore the functional significance of ETS1–BRD4 interaction, we performed ETS1 and BRD4 ChIP-seq in *ETS1* or *GFP* modRNA-transfected HUVEC cells. ETS1 and BRD4 chromatin occupancy generally correlated within promoter as well as distal regions (Fig. 3d, e, Supplementary Fig. 3b). Moreover, ETS1 overexpression increased ETS1 chromatin occupancy, with a corresponding increase in local BRD4 recruitment (Fig. 3d, e, Supplementary Fig. 3b). These data demonstrate that ETS1 directly interacts with BRD4, and that this interaction recruits BRD4 to ETS1-bound chromatin in ECs.

**ETS1 interacts with BRD4 through acetylation of its N-terminal domain.** We further analyzed the interaction between ETS1 and BRD4. ETS1 contains five annotated functional domains: the N-terminal (NT), pointed (PNT), acidic, exonVII, and ETS domains[22]. To map the BRD4-interacting domain of ETS1, we performed co-IP experiments between BRD4 and ETS1 mutants lacking selected functional domains (Fig. 3f, g). We found that the NT domain is necessary and sufficient for BRD4 binding, and that the PNT domain enhances this interaction.

Next, we mapped the region of BRD4 that binds ETS1. BRD4 has three known functional domains, two bromodomains (Bromo1 and Bromo2) and a long C-terminal domain that binds P-TEFb. In Co-IP assays, either Bromo1 or Bromo2, but not the C-terminal domain, bound ETS1 (Fig. 3h) or the ETS1 NT domain (Supplementary Fig. 3c).

Bromodomains bind acetylated lysine residues[23]. We therefore hypothesized that it binds acetylated lysines within the ETS1 NT domain. To map ETS1 acetylated lysine residues, we co-expressed Flag-tagged ETS1 and CBP in 293T cells. Mass spectroscopy of affinity purified ETS1 showed that lysine residues at positions 8 and 18, within the NT domain, were the most heavily acetylated (Supplementary Table 2). We confirmed these findings by mass spectroscopy of endogenous ETS1 immunoprecipitated from HUVEC cells (Supplementary Fig. 4a). These residues are highly conserved from zebrafish to humans (Fig. 4a). To further test if K8 and K18 acetylation is functionally important for ETS1–BRD4 interaction, we synthesized N-terminally biotinylated peptides and measured their binding to BRD4. BRD4 bound strongly to K8, 18 doubly acetylated NT peptide, less strongly to singly acetylated peptides, and weakly to unacetylated peptide (Fig. 4b). Consistent with this observation, substituting the acetyl-lysine mimetic residue glutamine for K8 and K18 (K8;18Q) increased BRD4 binding, while substituting acetylation resistant arginine (K8;18R) impaired it (Fig. 4c). Taken together, these results demonstrate that the ETS1 NT domain binds directly to BRD4, and K8;18 acetylation enhances this interaction.

**VEGF enhances ETS1–P-TEFb interaction through MAPK pathway.** ETS1 has been reported to bind to the lysine acetyl-transferase *CBP*, and this interaction was enhanced by ERK-dependent ETS1 phosphorylation on threonine and serine residues at *ETS1* positions 38 and 41 (T38 and S41)[24, 25]. To test whether CBP catalyzes K8;18 acetylation, we co-transfected CBP and ETS1 into 293T cells and detected ETS1 acetylation by using acetyl-lysine specific antibody. CBP increased acetylation of both full length ETS1 and ETS1-NT and stimulated their binding to BRD4 (Fig. 4d, e). The acetylation resistant mutation K8;18R abrogated CBP-induced acetylation, confirming that these residues are the major targets of CBP acetyltransferase activity. Remarkably, abrogating the ERK phosphorylation sites on ETS1 by T38A and S41A substitutions also blocked ETS1 acetylation (Fig. 4d). These data support a model in which ETS1 phosphorylation recruits CBP to bind ETS1 and acetylate it on K8 and K18, leading to ETS1 binding to BRD4.

We therefore asked whether VEGF regulates ETS1 phosphorylation at T38 in ECs and thereby regulates ETS1–CBP–BRD4 interaction. An ETS1-T38 phosphospecific antibody showed that VEGF treatment triggered ETS1-T38 phosphorylation (Fig. 4f, Supplementary Fig. 4a). Blocking the ERK pathway with selective inhibitor PD598059 abrogated VEGF-driven ETS1 T38 phosphorylation (Fig. 4g). In contrast, selective inhibition of calcium-calmodulin dependent kinase II (KN93) or SRC (Dasatinib), enzymes reported to phosphorylate ETS1[26, 27], did not significantly impact ETS1 T38 phosphorylation. VEGF increased acetylation of ETS1-NT (Fig. 4h). Moreover, VEGF-induced

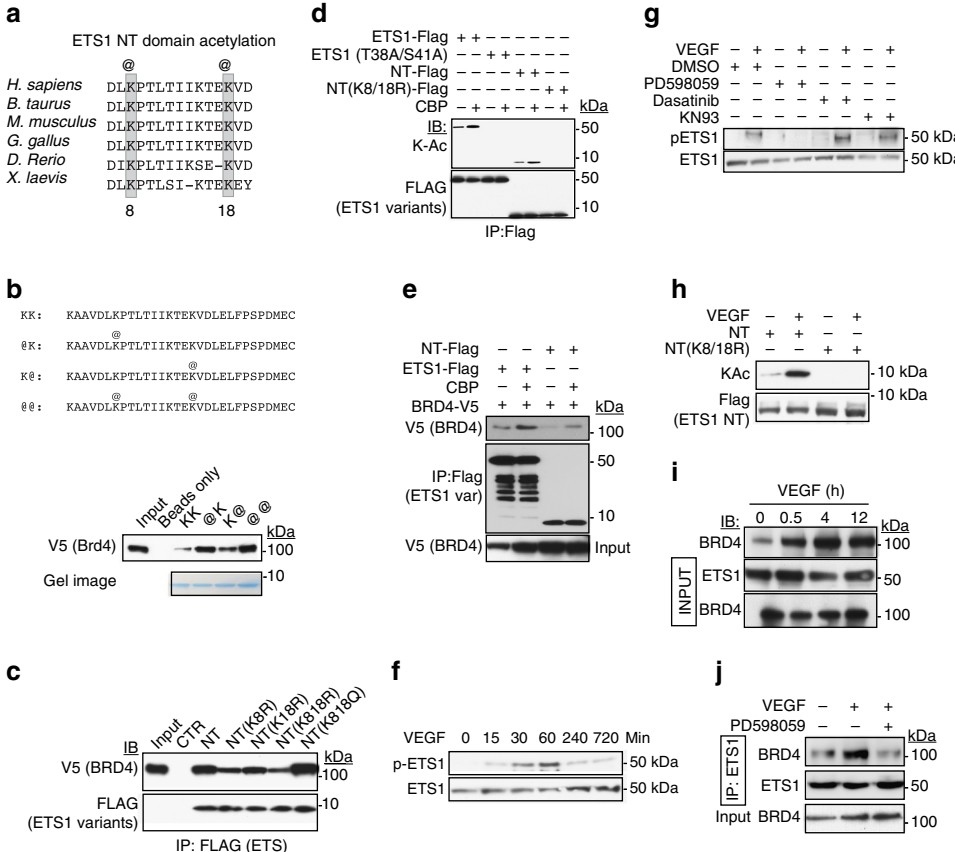

**Fig. 4** VEGF increased ETS1 acetylation and interaction with BRD4. **a** ClustalW alignment of a region within the ETS1 NT domain. K8 and K18 were major sites of acetylation, as determined by mass spectrometry. **b** ETS1 NT acetylation enhances BRD4 binding. Peptides corresponding to the N-terminus of ETS1 were synthesized with either lysine or acetyl-lysine (@) at positions 8 and 18. Immobilized peptides were incubated with V5-BRD4-containing cell lysates. K8 or K18 acetylation enhanced BRD4 binding. **c** ETS1–BRD4 interaction is modulated by K8 and K18 acetylation. The ETS1 NT domain was expressed with mutations that abrogate (R) or mimic (Q) lysine acetylation. Interaction with co-transfected V5-BRD4 was assessed by co-immunoprecipitation. **d** CBP acetylates ETS1 at K8 and K18. ETS1 acetylation was assessed by immunoprecipitation followed by immunoblotting with acetyl-lysine specific antibody. ETS1 acetylation required K8 and K18, and ERK phosphorylation sites T38 and S41. **e** CBP stimulates BRD4 binding to ETS1 NT domain. Expression constructs were co-transfected into 293T cells. BRD4 co-precipitated by ETS1 was detected by immunoblotting. CBP stimulated ETS1–BRD4 interaction. **f** VEGF stimulates ETS1 phosphorylation. HUVEC cells were treated with 50 ng ml$^{-1}$ VEGF for the indicated time. ETS1 T38 phosphorylation (p-ETS1) was detected using specific antibody. **g** ERK is required to phosphorylate ETS1 downstream of VEGF. ERK pathway inhibitor PD598059 blocked ETS1 phosphorylation in HUVECs stimulated by VEGF, but inhibitors of other ETS1 kinases (KN93 or Dasatinib) did not. **h** VEGF induced ETS1 NT acetylation at K8 and K18. In HUVEC cells, VEGF treatment induced robust acetylation of the ETS1 NT domain, but this was abolished by K8;18R mutation. **i** VEGF-induced ETS1–BRD4 interaction. ETS1 was immunoprecipitated from HUVEC cells treated with VEGF for the indicated time. Co-precipitated BRD4 was measured by immunoblotting. **j** VEGF-induced ETS1–BRD4 interaction requires ERK activity. Co-immunoprecipitation assay as in **i** was performed with or without ERK inhibitor PD598059

acetylation was blocked by T38A; S41A mutation (Fig. 4h, Supplementary Fig. 4a, b), indicating that phosphorylation of these residues is required for K8;18 acetylation induced by VEGF. VEGF stimulation also increased ETS1–BRD4 interaction in HUVEC, and this was blocked by ERK inhibition (Fig. 4i, j). Together, these results demonstrate that VEGF stimulates the interaction of ETS1 and BRD4 by activating ERK, which phosphorylates ETS1, leading to CBP recruitment and ETS1 acetylation.

**ETS1 regulates VEGF downstream transcription.** Reports from our group and others have implicated ETS1 in VEGF-dependent transcriptional responses[3, 28]. Here, we found that VEGF increased ETS1–BRD4 interaction and stimulated release of paused RNAPII. To better delineate the role of ETS1 in mediating VEGF downstream transcription, we analyzed ETS1 chromatin occupancy in VEGF-stimulated HUVECs by ChIP-seq[3] (Fig. 1a). VEGF increased the number of ETS1 regions from 17,585 at 0 h

to 35,253 and 37,269 at 4 and 12 h, respectively (Fig. 5a). Most of these new binding events were in the vicinity of the same genes, since the genes neighboring ETS1 regions remained largely unchanged throughout the time course (Fig. 5b). In addition, ETS1 signal at occupied sites increased genome-wide at 4 and 12 h (Fig. 5c-d). These increases in chromatin occupancy were likely due to VEGF-induced increases in ETS1 DNA binding affinity[28] rather than to upregulation of ETS1 protein, since ETS1 levels were stable over this time course (Fig. 4f). Increased ETS1 chromatin occupancy correlated with increased transcription of neighboring genes at 4 and 12 h (Fig. 5e). These data suggest that changes in ETS1 chromatin occupancy are a globally important mechanism for VEGF downstream transcriptional regulation, especially at later time points.

To further test the function of ETS1 and ETS1-stimulated RNAPII pause release in VEGF transcriptional regulation, we stably over-expressed ETS1-NT, ETS1-NT-K8;18R, ETS1-NT-K8;18Q, or GFP (control) in HUVECs using lentivirus

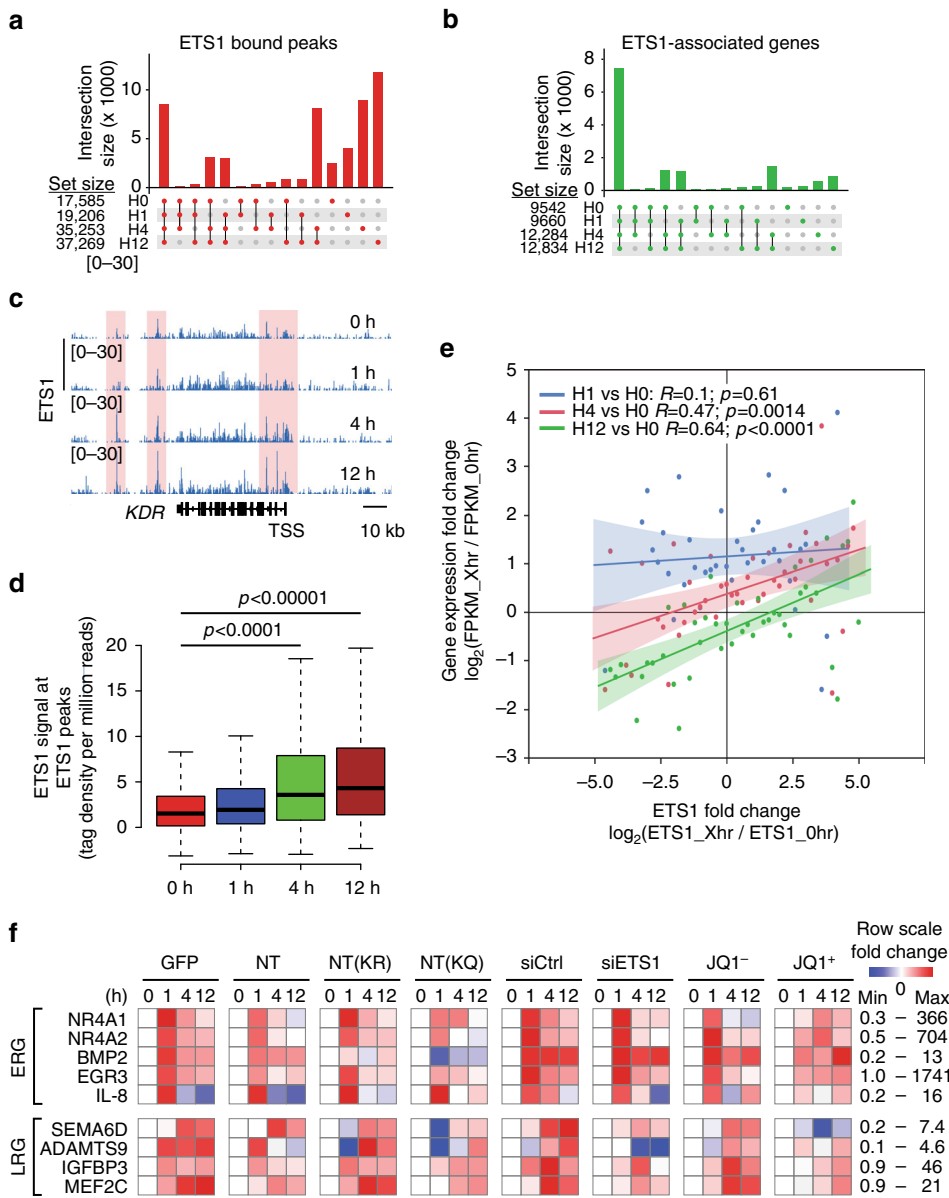

**Fig. 5** ETS1 amplified VEGF downstream transcription. **a**, **b** UpSet plots[45] showing the overlap of ETS1-bound peaks (**a**) or ETS1-associated genes (**b**) between the four time points (0, 1, 4, 12 h). The *bar* represents the number of peaks shared by the time points indicated by the *colored dots* and not by the time points indicated by the *gray dots*. Most ETS1 peaks were not shared between time points, but most ETS1-associated genes were shared. **c** VEGF stimulated ETS1 chromatin occupancy. Genome browser view of ETS1 occupancy at the *KDR* locus at the indicated times of VEGF stimulation. *Pink* highlights regions with greater VEGF occupancy over the time course. **d** Genome-wide gain of ETS1 signal at ETS1-bound regions during VEGF stimulation time course. ETS1 binding increased genome-wide at 4 and 12 h. Mann–Whitney *U*-test. *Dark line* and *boxes* represent the median and 25th and 75th percentiles. The *whiskers* represent median ± 1.5 times the interquartile range. **e**. VEGF-induced changes in gene expression correlated to changes in ETS1 promoter occupancy. Analysis was limited to expressed genes with ETS1 promoter occupancy. Expression and ETS1 signal at the indicated time point of VEGF treatment are expressed as fold change compared to hour 0. Each point was calculated by grouping genes with similar ETS1 occupancy change. Correlation and *p*-values were plotted using a linear regression model. **f** VEGF-induced gene activation requires ETS1–BRD4 interaction. HUVEC cells were transduced with lentivirus expressing the indicated proteins, transfected with siRNA, or treated with small molecule JQ1. Gene expression after the indicated number of hours of VEGF stimulation was measured by qRT-PCR and displayed as a heat map. Log10 fold-change compared to time 0 was row scaled. Linear minimum and maximum fold-change values for each row are listed on the *right*

(Supplementary Fig. 5a, b). By inhibiting endogenous ETS1–BRD4 interaction, we anticipated that ETS1-NT would have dominant negative activity. Since NT-K8;18R (abbreviated NT(KR)) reduced BRD4 affinity, we expected that this mutation would attenuate dominant negative activity. Similarly, NT-K8;18Q (abbreviated NT(KQ)) enhanced BRD4 affinity, leading us to expect that this mutation would enhance dominant negative activity. HUVEC expressing these peptides were treated with VEGF, and the

expression of several early response genes (ERGs; peak at 1 h) and late response genes (LRGs; highest at 4 or 12 h; Fig. 5f) was measured by quantitative RT-PCR. The ETS1-NT domain interfered with expression of most of the early genes at 1 h and late response genes at 4 and 12 h. As expected, this effect was attenuated by acetylation-refractory mutant NT(KR) and enhanced by acetylation-mimetic mutant NT(KQ). These data support our model that ETS1–BRD4 interaction through VEGF-

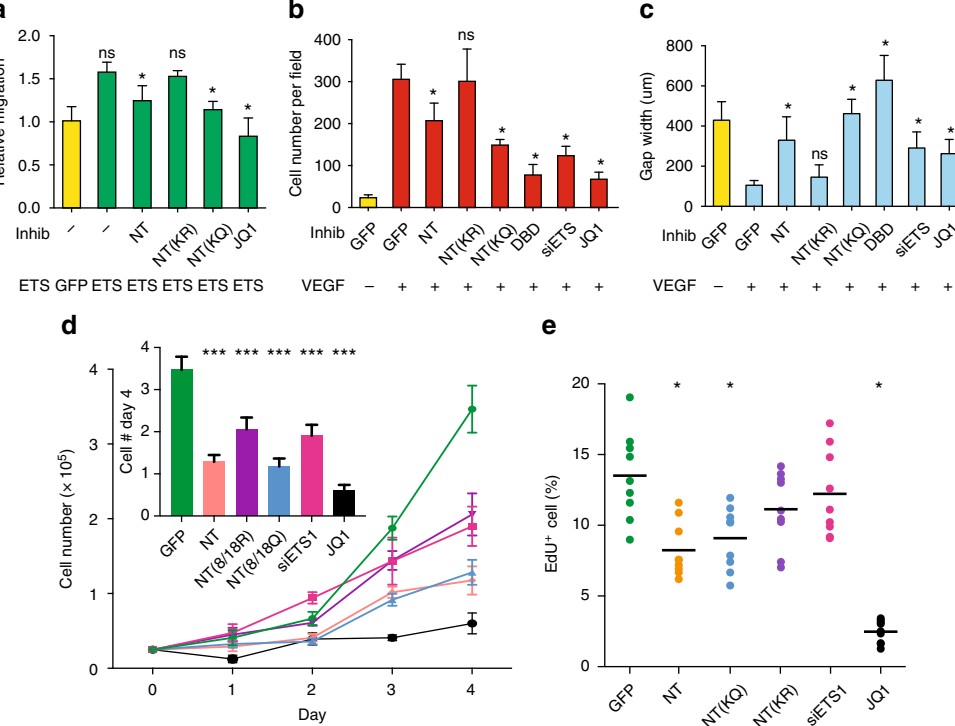

**Fig. 6** ETS1–BRD4 interaction is required for VEGF-driven angiogenic responses in vitro. **a** ETS1-driven EC migration was inhibited by interfering with BRD4-mediated RNAPII pause release or ETS1–BRD4 interaction. HUVEC cells were transduced with the indicated lentivirus, or treated with JQ1, and migration was measured using a trans-well assay. ETS1 stimulated migration, and this was attenuated by peptides that block ETS1–BRD4 interaction or by JQ1 inhibition of BRD4. **b** VEGF-driven EC migration was inhibited by interfering with BRD4-mediated RNAPII pause release or ETS-BRD4 interaction. Trans well assay was performed as in **a**. VEGF was used at 30 ng ml$^{-1}$. Student's t test vs. VEGF-stimulated cells without inhibitor: *$p < 0.05$. ns: not significant, n = 4. Bar = 250 μm. **c** VEGF-driven wound healing of HUVEC monolayer was inhibited by interfering with BRD4-mediated RNAPII pause release or ETS1–BRD4 interaction. The width of the scratch at the end of the culture period is inversely related to EC migration capacity. Student's t-test vs. VEGF-treated cells without inhibitor: *$p < 0.05$. n = 3. ns, not significant. **d, e** HUVEC proliferation requires ETS1–BRD4 interaction and BRD4 activity. HUVECs were transduced with lentivirus expressing the indicated proteins, transfected with siRNA, or treated with JQ1. Cells were cultured in EGM2 growth medium. Cell number was counted daily **c**, or traversal of S phase was measured by culturing cells in the nucleotide analog EdU (**e**). Cell counts at day 4 are plotted in the inset in **c**. Bar graphs show mean ± s.d. Intergroup comparisons were made with Student's t-test vs. baseline (yellow, **a-c**; green, **d, e**): *$p < 0.05$; ***$p < 0.001$; ns, not significant. For each experiment, n = 4

dependent ETS1 NT domain acetylation regulates EC transcriptional responses.

We also used siRNA and small molecules to deplete ETS1 or inhibit BRD4, respectively, in this system (Fig. 5f, right four panels). Inhibiting BRD4 using JQ1 dramatically blocked VEGF stimulation of both ERGs and LRGs, compared to vehicle control. This result is consistent with our prior work, which showed that VEGF stimulation of HUVECs reduces RNAPII pausing at early response genes[13] and points to a functionally important role for release of paused RNAPII in rapid VEGF-induced transcriptional changes. In contrast, ETS1 siRNA primarily blunted upregulation of the late response genes at 4 and 12 h (4 of 4 tested), whereas it had relatively weak effect on the early response genes at 1 h (Fig. 5f). This result supports the critical, non-redundant role of ETS1 in activating LRGs downstream of VEGF, consistent with the correlation between increased ETS1 chromatin occupancy and gene expression at these later time points (Fig. 5f). The minimal effect of ETS1 knockdown on ERG activation may be attributable to incomplete knockdown or to ETS1 functional redundancy for activation of these genes, consistent with known functional overlap between ETS1 and ETS2[5] and the presence of additional ETS family TFs in these cells.

Collectively, our studies show that VEGF induces ETS1 acetylation on K8 and K18 to stimulate BRD4 interaction and downstream transcriptional activation.

**RNAPII pausing-release promotes in vitro angiogenesis.** The observation that ETS1-mediated RNAPII pausing-release strongly regulated VEGF downstream transcription prompted us to evaluate its role in angiogenesis. ETS1 has been reported to be a strong inducer of endothelial migration[4]. HUVECs that stably over-expressed ETS1 exhibited greater migration in transwell assays than GFP-expressing controls (Fig. 6a, Supplementary Fig. 5c). This enhancement was negated by JQ1 inhibition of BRD4, implicating BRD4-mediated pause release in this ETS1 gain-of-function phenotype. ETS1 enhancement was dampened by stable expression of the ETS1-NT peptide and the acetylation mimetic mutant ETS1-NT(KQ). In contrast, stable expression of the acetylation deficient mutant ETS1-NT(KR) had no effect. These data indicate that ETS1 stimulation of EC migration depends upon its interactions with BRD4 via acetylation at K8 and K18.

We further probed the role of ETS1–BRD4 interaction in VEGF-induced EC migration, using both the transwell assay (Fig. 6b, Supplementary Fig. 5d) and the wound healing assay (Fig. 6c, Supplementary Fig. 5e). VEGF stimulation markedly increased HUVEC migration, and this effect was blocked by both ETS1 depletion with siRNA and BRD4 inhibition by JQ1 (Fig. 6b, Supplementary Fig. 5d). We further investigated the requirement for ETS1–BRD4 interaction by stably expressing ETS1-NT domain mutants. ETS1-NT attenuated the effect of VEGF. The

acetylation mimetic peptide *ETS1*-NT(KQ) had a stronger dominant negative effect, while the acetylation-deficient peptide *ETS1*-NT(KR) did not have a significant inhibitory effect. These data indicate that *ETS1* stimulation of EC migration depends upon its interaction with *BRD4* via acetylation at K8 and K18. The *ETS1* DNA-binding domain (DBD), previously shown to have dominant negative activity[25], also blocked EC migration,

underscoring the importance of ETS1 in this process. Together, these results indicate that RNAPII pause release and ETS1–BRD4 interaction is functionally relevant for EC migration in response to VEGF.

Stimulation of EC proliferation is another in vitro hallmark of VEGF angiogenic activity. Therefore, we measured the proliferative response of HUVECs to *VEGF* by measuring both cell

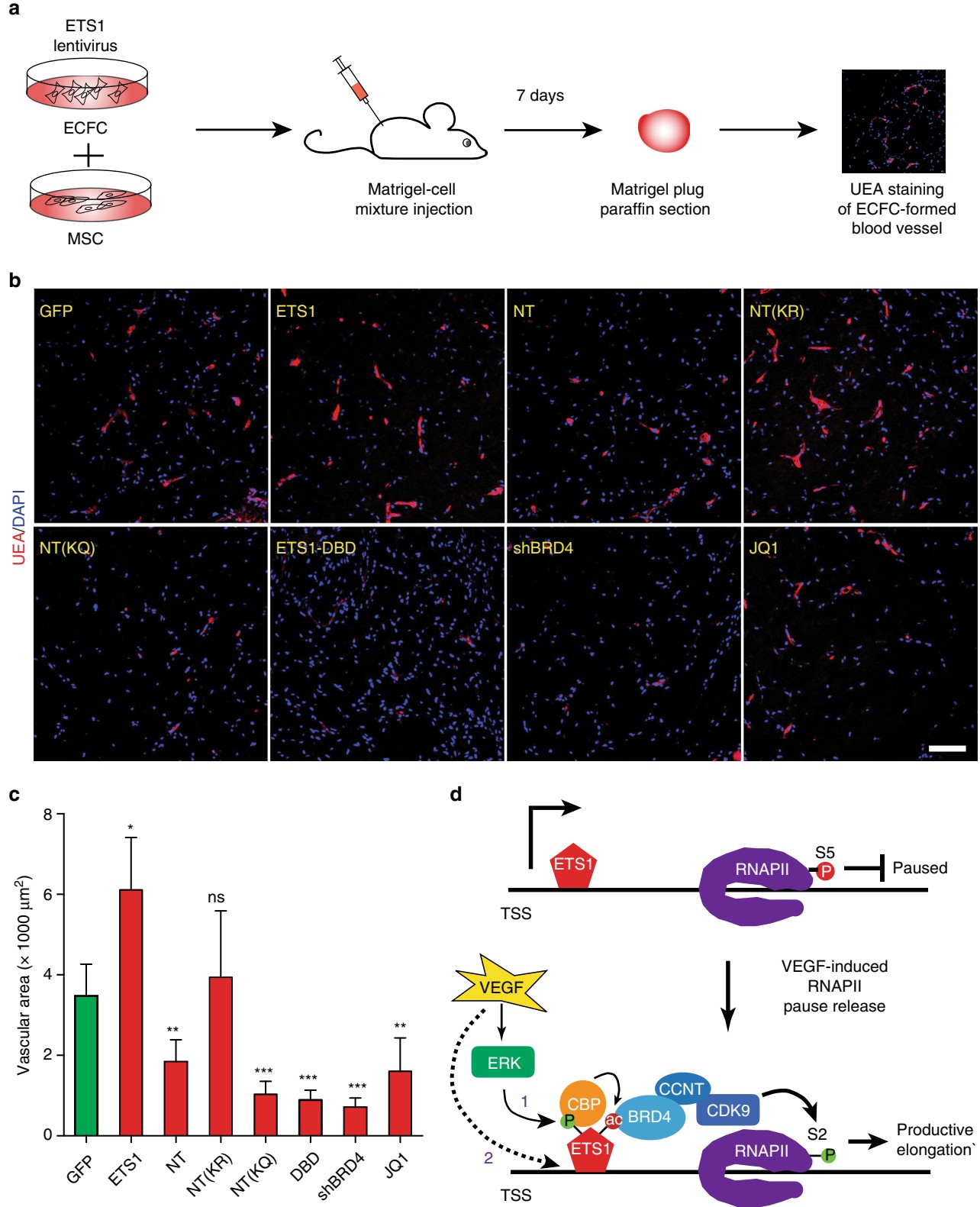

number (Fig. 6d) and transit through S-phase (percent of cells that incorporated the nucleotide analog EdU, Fig. 6e). In both assays, BRD4 inhibition by JQ1 powerfully blocked HUVEC proliferation induced by VEGF, consistent with its effect on many cancer cell lines. ETS1-NT and ETS1-NT(KQ) both strongly inhibited proliferation, whereas ETS1-NT(KR) had attenuated dominant negative activity. ETS1 depletion by siRNA also decreased proliferation, although its effect was weaker than the dominant negative peptides, possibly due to functional redundancy as well as incomplete knockdown.

Together, these results highlight the importance of RNAPII pause release and its modulation by ETS1–BRD4 interaction for in vitro EC angiogenic responses to *VEGF*.

**RNAPII pausing-release promotes in vivo angiogenesis**. To test the hypothesis that ETS1–BRD4 interaction mediates angiogenesis in vivo, we used the matrigel plug assay to measure neovascular network formation by human endothelial colony forming cells (ECFCs), a type of endothelial progenitor cell that robustly forms vessels in this assay (Fig. 7a and Supplementary Fig. 6a; see methods)[29]. ECFCs expressed ETS1, BRD4, and RNAPII-pS2 at levels comparable to HUVECs (Supplementary Fig. 6b). Lentiviral *ETS1* over-expression strongly stimulated formation of vessels, marked by Ulex europaeus agglutinin (UEA), while lentiviral ETS1-NT expression significant inhibited it. Again, the KQ mutation potentiated dominant negative activity, whereas the KR mutation abrogated it. Lentiviral expression of ETS1 DNA binding domain (DBD) also inhibited angiogenesis in this assay, consistent with its dominant negative activity (Fig. 6b, c and ref. [25]). BRD4 inhibition by JQ1 or lentivirally expressed shRNA (Fig. 7b, c, Supplementary Fig. 6c) also strongly antagonized vessel formation, underscoring the critical role of BRD4 and RNAPII pause release in angiogenesis in vivo. Increased EC apoptosis and decreased EC proliferation contributed to defective angiogenesis by NT, NT(KQ), and shBRD4-treated ECFCs (Supplementary Fig. 6c–f). These results were consistent with our in vitro angiogenesis assays and suggest that ETS1–BRD4 interaction through K8 and K18 acetylation are indispensable for in vivo angiogenesis.

Hypoxia-induced VEGF secretion is a major contributor to the pathogenesis of retinopathy of prematurity, in which aberrant angiogenesis damages the retina of premature infants. *ETS1* mRNA was upregulated in the oxygen-induced retinopathy (OIR) mouse model of this disease[30, 31] (Supplementary Fig. 7a), suggesting that ETS1 may participate in the pathogenesis of retinopathy. JQ1 inhibition of RNAPII pause release reduced the extent of pathological neovascularization (Supplementary Fig. 7b, c). Moreover, intra-retinal injection of lentivirus expressing dominant negative ETS1 NT peptides reduced pathological neovascularization compared to control (GFP; Supplementary Fig. 7d, e). Again, the KQ mutation increased potency, while the KR mutation reduced it.

Together, these experiments suggest that ETS1–BRD4 interaction is functionally significant for angiogenesis in vivo.

## Discussion

RNAPII pausing and pausing-release is the rate-limiting step for productive transcription of many genes[7, 8]. Here we show that regulation of RNAPII pause release is critical for angiogenesis, and we delineate a molecular pathway that links VEGF to broad induction of EC gene transcription through RNAPII pause release (Fig. 7d). VEGF activates ERK, which phosphorylates ETS1 at T38 and S41. CBP is recruited to phosphorylated ETS1, inducing acetylation of ETS1 and likely other local chromatin elements such as histones. Acetylated ETS1 recruits BRD4 and the active P-TEFb pause release complex, thereby rapidly and widely increasing gene expression. In addition, VEGF increases ETS1 chromatin occupancy, which contributes to upregulation of late response genes, likely through both increased RNAPII recruitment and pause release (Fig. 7d).

Recently, a new class of transcription factors has been identified that broadly regulates gene expression by promoting RNAPII pause release[8, 10, 11]. These "transcriptional amplifiers", such as MYC, the prototypical factor in this class, exhibit the following characteristics: (1) occupancy of the promoters of most expressed genes; (2) correlation between promoter occupancy, RNAPII occupancy, and gene expression; and (3) broad stimulation of RNAPII pause release with resulting broad increase in transcript abundance. We found that ETS1 shares these characteristics, indicating that it is a transcriptional amplifier that enhances productive RNAPII elongation in ECs and possibly other cell types in which it is expressed, including blood and prostate cancer cells. Our data indicate that ETS1 co-occupies promoters with MYC. Future studies will determine whether these factors act synergistically or independently to stimulate RNAPII pause release.

While ETS1 and MYC share many properties, they also have interesting differences. MYC upregulated total RNA, as it increased rRNAs and tRNAs in addition to mRNAs[11]. However, ETS1 amplified mRNAs but not rRNA, suggesting that ETS1 more selectively impacts RNAPII transcription rather than transcription by other RNA polymerases. Another difference is that MYC is broadly expressed in many cell types, whereas ETS1 is restricted to select cell types, most notably endothelial and hematopoietic lineages. Thus, ETS1 is an example of a cell-type restricted transcriptional amplifier. Finally, MYC transcription is dependent upon BRD4-mediated RNAPII pause release, making MYC expression susceptible to inhibition by JQ1[32], the small molecule inhibitor of BRD4, and suggesting a potential positive feedback loop in which MYC induces its own expression through release of paused RNAPII. ETS1 expression, on the other hand, was not affected by JQ1, indicating that this type of positive feedback loop is not a general feature of transcriptional amplifiers. The relationship between RNAPII pause release and MYC, and co-occupancy between MYC and ETS1, suggests possible complex regulatory circuits which are not disentangled by our current study and which will be a fruitful area for further investigation.

Promoter proximal RNAPII pausing has been suggested to regulate genes responsive to environmental stress, immunological

**Fig. 7** ETS1–P-TEFb modulated in vitro angiogenesis. **a** Experimental design for the matrigel plug assay. ECFCs were transduced with lentivirus expressing indicated proteins or shRNA. Alternatively, ECFCs were cultured in JQ1 overnight, prior to matrigel implantation. Cells were then mixed with MSCs in matrigel, and injected subcutaneously into mice. **b** Matrigel plugs were sectioned and stained with fluorescently labeled UEA, which binds human ECs. Representative confocal images are shown. *Bar* = 100 μm. **c** Inhibition of ETS1 activity, BRD4 activity, or ETS1–BRD4 interaction impaired vessel formation in the matrigel plug assay. UEA-stained vascular area in matrigel plugs was quantified. Student's *t*-test compared to GFP control: *$p < 0.05$; **$p < 0.01$; ***$p < 0.001$. $n = 3$–5. *Bar plots* show mean ± s.d. **d** Working model of ETS1-mediated transcriptional response to VEGF. VEGF activates ERK, which phosphorylates ETS1 (1). This recruits CBP to ETS1. CBP acetylates ETS1 at K8,18 in the NT domain, stimulating BRD4 binding and activation of P-TEFb. P-TEFb phosphorylates serine 2 within the C-terminal domain of RNAPII and releases RNAPII pausing, resulting in productive elongation. VEGF also stimulates ETS1 chromatin occupancy (2), enhancing its effect on both RNAPII initiation and pause release

stimulation and developmental cues. Previously, we and others showed that VEGF promoted RNAPII pausing release on a genome-wide scale[13,33]. However, the molecular pathways linking these events to RNAPII pause release are not well understood. In this study, we identified two mechanisms by which VEGF influences RNAPII pause release through ETS1. First, VEGF activates the MAPK pathway, and ERK, a major effector of this signaling pathway, phosphorylates ETS1 at T38 and S41[25], resulting in ETS1 recruitment of CBP, a lysine acetyltransferase[25]. CBP then acetylates ETS1 at K8 and K18, enhancing ETS1–BRD4 binding. This interaction recruits P-TEFb to ETS1-occupied chromatin sites, which are highly enriched within promoter regions of expressed genes, positioning P-TEFb to stimulate release of promoter-proximal paused RNAPII. BRD4 has also been reported to be an atypical kinase that itself phosphorylates RNAPII-S2[20], suggesting an additional mechanism by which ETS1–BRD4 stimulates RNAPII pause release.

Second, at later time points VEGF stimulated ETS1 chromatin occupancy genome-wide without altering overall ETS1 protein level, and this correlated with gene upregulation. Increased ETS1 chromatin occupancy could directly recruit BRD4 to stimulate RNAPII pause release. Alternatively, increased ETS1 chromatin occupancy could recruit CBP to enhance local acetylation of histones and other proteins, thereby stimulating traditional enhancer activity or BRD4-mediated RNAPII pause release. We observed increased ETS1 chromatin occupancy primarily at later time points (hours 4 and 12). However, ETS1–BRD4-mediated stimulation of RNAPII pause release operated as early as 1 h. One interpretation of these data is that ETS1 phosphorylation and acetylation of pre-bound ETS1 leads to rapid BRD4 recruitment and release of paused RNAPII, resulting in rapid upregulation of ERGs. At later time points, VEGF-stimulated ETS1 chromatin occupancy contributes to upregulation of LRGs, potentially through both traditional transcriptional enhancers that increase RNAPII initiation and through enhanced RNAPII pause release.

Our experiments suggest that manipulation of this VEGF–ETS1–BRD4 transcriptional regulatory pathway may be a new avenue to ameliorate retinopathy, cancer, and other diseases of excessive angiogenesis. However, further studies will be required to more comprehensively understand the in vivo significance of VEGF regulation of RNAPII pause release for vessel formation and maintenance. On the other hand, our study also highlights the need to be alert for adverse effects of chronic angiogenic suppression that may result from BRD4 inhibition for treatment of other diseases. In this regard, the ETS1–BRD4 interaction may represent a more selective therapeutic target than BRD4 itself, with potential applications to ameliorate diseases caused by excessive angiogenesis, such as retinopathy of prematurity.

## Methods

**Mice**. Male nude mice (CByJ.Cg-Foxn1[nu]/J, Jax #000711) at the age of 6–8 weeks from JAX were used for Matrigel Plug Assay. Wildtype C57BL6 mice of both genders were used for the OIR experiment at P7–P17. All experiments with mice were performed under protocols approved by the Institutional Animal Care and Use Committee of Shanghai Jiao Tong University and Boston Children's Hospital.

The matrigel plug assay was performed as described[29]. Human Endothelial Clone Forming Cells (ECFCs; passages 8–12) were transduced with lentivirus for 48 h and then collected by trypsinization. 0.8 million ECFC and 2 million human mesenchymal stem cells (MSC, Lonza) were mixed with 200 μl ice-cold matrigel (Corning, 356237) and injected under the abdominal skin of nude mice. After 7 days, the gel plugs were dissected out and fixed in 4% paraformaldehyde overnight and preserved in paraffin. Paraffin sections were incubated for 30 min in boiled antigen retrieval buffer (sodium citrate, 2 mg ml$^{-1}$, pH 6.0; Tween 0.05%) and then blocked with 1% BSA plus 10% serum that matched the host species of the second antibody (Alexa488-Donkey anti-Rabbit, Thermo Fisher). Cell proliferation was measured using Ki-67 antibody (1: 200, Neomarker). Cell apoptosis was measured using anti-activated Caspase 3 antibody (1:200, Millipore). Blood vessels formed were stained by UEA (1:200, Vector Labs), a lectin that only recognizes human ECs. Proliferative or apoptotic cell numbers and vascular area were measured using ImageJ.

Oxygen induced retinopathy (OIR) was performed as described[30, 31]. Briefly, neonatal mice (129S) were exposed to 75% oxygen from postnatal day 7 to 12, along with their nursing mothers. At postnatal day 12, mice were returned to room air. Where indicated, 0.5 μl lentivirus (10$^7$ per eye) were intravitreously injected into the retina of anesthetized pups at postnatal day 1, or JQ1 (5 mg kg$^{-1}$ in 10% hydroxypropyl beta-cyclodextrin) was delivered to the mice daily by IP injection from postnatal day 12 to 16. At postnatal day 17 the mice were anesthetized, retinas were dissected and The retina was whole-mount stained with Alexa Fluor 594 isolectin GS-IB4 (1:200, Thermo Fisher), which labels ECs. Retinas were imaged using a Zeiss AxioObserver.Z1 microscope and AxioCam MRm camera. QPathological neovascularization was quantified as a percentage of total retinal areas performed using ImageJ with the SWIFT NV plug-in[32].

**Cell culture**. Primary HUVEC cells at 3–6 passages were purchased from Lonza (CC-2519) and grown in EGM2 full medium (CC-3162, Lonza) for regular maintenance. Prior to VEGF stimulation, cells were cultured overnight in basal culture medium (EBM2 plus 1% FBS). The hour 0 sample was collected just prior to addition of 50 ng ml$^{-1}$ VEGF (R&D Systems). Stimulated cells were collected at 1, 4, and 12 h for ChiP-seq or RNA-seq experiments. JQ1 (cat. no. 4499, Tocris Bioscience) was used at 500 nM.

HUVEC cells were transfected with siRNAs (10 ng ml$^{-1}$, Qiagen; sequences in Table S3) using RNAiMAX (Life Technologies) following the manufacturer's protocol. Cells were analyzed 24–48 h after transfection.

**In vitro migration assay**. HUVEC in vitro transwell assay was performed as described[36, 37]. Briefly, 2–5 × 10$^4$ HUVEC cells were transduced with lentivirus for 2 days, collected by trypsinization, diluted into migration medium (EBM2 + 1% FBS), and then loaded onto the upper side of the transwell. 30 ng ml$^{-1}$ VEGF in 600 μl migration medium was added to the lower chamber to induce migration. Migrated cells were stained with crystal violet (0.09% crystal violet in 10% ethanol) and counted with the "Particle Analysis" function of ImageJ.

**Lentivirus production and transduction**. ETS1 and domain deletion variants fused with 3×Flag were cloned into the NheI site of Phage-CMV-zsGreen vectors by Gibson assembly. Where indicated, a nuclear location sequence (NLS) was added to ensure nuclear entry. To make ETS1 and BRD4 knock down constructs, we used PLKO.1-Puro to express ETS1 and BRD4 specific shRNA. These vectors were transfected to 293 T cells together with VSV-G expressing envelope vector (PDM2.G) and packaging vector (psPAX2) to generate lentivirus particles. The virus particles were purified and concentrated with PEG6000 from cell culture medium and titered with Lenti-X p24 Rapid Titer Kit (Clontech).

HUVEC or ECFCs were transduced with lentivirus at one to two multiplicity of infection in the presence of 8 μg ml$^{-1}$ of hexadimethrine bromide. The virus-transducing medium was replaced with fresh EGM2 medium 4 h after infection. The transfection efficiency was monitored by zsGreen expression to ensure that infection efficiency each time was >90%.

**Modified RNA generation and transfection**. ETS1 and GFP protein coding sequences were cloned into a TOPO vector with T7 promoter and 5′ UTR (Gift from Jian Ding & Da-Zhi Wang) and then PCR amplified using Phusion high fidelity polymerase (New England Biolabs) to generate a linear DNA template containing the T7 promoter, 5′UTR and polyT for in vitro transcription (see Table S3 for primer sequences). The in vitro transcription was performed with the MEGAscript Kit (Ambion) in accordance with the manufacturer's protocol, except that a custom NTP mix was used. This mix was composed of 2.4 nM 3′-0-Me-m7G (50)ppp(50)G ARCA cap analog (New England Biolabs), ATP (USB), GTP (USB), 5-methylcytidine triphosphate (TriLink), and pseudouridine triphosphate (Tri-Link). RNA transcripts were digested with DNaseI and then treated with Antarctic Phosphatase (New England Biolabs) to remove 5′ triphosphates. The final transcribed RNA was purified using the Megaclear kit (Themo Fisher) and adjusted to 100 ng μl$^{-1}$ with TE buffer.

For modRNA transfection, B18R (4 μg ml$^{-1}$) was added to HUVEC cells for 2 h to limit type I interferon activation. One well of a six well dish was transfected with 500 ng of modRNA using Lipofectamine RNAiMAX. GFP and ETS1 expression were examined by fluorescent microscopy and Western blotting 12 h after transfection.

**Total mRNA detection**. Total RNA from 2 × 10$^5$ HUVECs was purified using the Qiagen RNeasy mini kit. Contaminating DNA was eliminated by on-column DNaseI digestion for 15 min. Extracted RNA was applied to TapeStation 2200 RNA ScreenTapes to evaluate the total RNA concentration and quality. An aliquot of 2 μl of 1:10 diluted ERRC spike-in RNA (Thermo Fisher) was added to each sample of extracted RNA. mRNA was selected using the Dynabeads mRNA Purification Kit (Life Technologies). The cleavage, adaptor-tagging, and reverse transcription of purified mRNA was performed with Scriptseq V2 RNA library kit (Epicentre). The concentration of tagged-mRNA and spike-in RNA was measured by real-time PCR

using primers complementary to library adapters and Sybr Green chemistry. RNA-seq libraries were sequenced on an Illumina Hiseq 2500 (50 nt paired end reads).

**Protein expression and interaction**. Protein expression was measured by western blotting. Uncropped western blot images are shown in Supplementary Fig. 8. Nuclear protein was extracted with High Salt Buffer (20 mM HEPES, 10 mM KCl, 1 mM EDTA, 1 mM DTT, 350 mM NaCl, 1% NP40, 0.1% SDS, 1 mM PMSF, 10 mM NaF), separated by 4–12% gradient sodium dodecyl sulphate-polyacrylamide gel electrophoresis (SDS-PAGE), and probed with primary antibodies (Santa Cruz: ETS1 SC-350, 1:1000; CDK9 SC-281, 1:1000; Abcam: RNAPIIS2 ab5095, 1:1000; ETS1 (pT38) ab59179 1:500; Millipore: RNAPII 05-623, 1:2000; Bethyl Laboratories: BRD4 A301-985A50, 1:2000; Cell signaling: HEXIM1 12604S, 1:1000; ELL2 and AFF 1:1000, gifts from Ali Shilatifard[38]).

For co-IP experiments in 293T cells (ATCC), expression constructs were cotransfected into 293 T cells with PEI (Polybiosciences, Inc.). Total protein was extracted with IP buffer (20 mM Tris-Cl pH 8, 150 mM NaCl, 1% NP-40, 1 mM EDTA, 15 U DNaseI (New England Biolabs), 10 mM sodium butyrate, 50 µg ml$^{-1}$ PMSF, 1× Protease Inhibitor Cocktail (Roche)) and then incubated with 30 µl M2 Flag antibody magnetic beads (Sigma) overnight at 4 °C. For IP experiments in HUVECs, nuclear extract was incubated with antigen-specific antibody (1 µg per 10 mg input protein) and then pulled down by Protein G Dynal beads (Thermo Fisher).

For detection of ETS1 acetylation sites, after pulldown with 100 µl M2 Flag antibody beads, ETS1-3×Flag was eluted with Flag peptide (5 mg ml$^{-1}$, Sigma) for 1 h with shaking at 4 °C. Eluted proteins were resolved by 10% SDS PAGE and stained using SimplyBlue SafeStain (Thermo Fisher). Gel bands were excised, cut into ~1 mm$^3$ pieces, and digested with 50 mM ammonium bicarbonate solution containing 12.5 ng µl$^{-1}$ modified sequencing-grade trypsin (Promega) at 4 °C. Peptides were later extracted by 50 mM ammonium bicarbonate solution and followed by one wash with a solution containing 50% acetonitrile and 1% formic acid. Samples were further resolved in a nano-scale reverse-phase HPLC capillary column and analyzed using a LTQ Orbitrap Velos Pro ion-trap mass spectrometer (Thermo Fisher) to generate a tandem mass spectrum of specific fragment ions for each peptide. Peptide identity was determined using the software package Sequest (Thermo Fisher). Data were filtered to between a one and two percent false discovery rate.

For peptide affinity pulldown assays, V5-tagged BRD4 expression construct was transfected into 293T cells. Whole cell extracts were prepared in IP buffer 2 days after transfection and incubated with 2 µg of biotin-labeled peptide (synthesized by LifeTein) at 4 °C for 2 h. Peptide and bound proteins were pulled down with streptavidin Dynal Beads for 2 h. The Dynal beads were washed four times with IP buffer and then eluted with 1× SDS Laemmli Buffer.

For the in vitro pulldown assay, His-ETS1 was cloned into pET32b vectors and expressed in BL21 bacterial by induction with 0.1 µM IPTG (Sigma). His-ETS1 was purified with immobilized cobalt affinity resin (Clontech). TNT Quick Coupled Transcription/Translation System was deployed to express Flag-BRD4. Purified His-ETS1 was incubated with in vitro translated Flag-BRD4 overnight at 4 °C. After the pull down with cobalt resin, interacting BRD4 were detected by immunoblot with M2 Flag monoclonal antibody (1:5000, Sigma).

**ChiP-seq**. ChiP-seq was performed as described previously with minor modifications. $2 \times 10^7$ cells (ChIP of modified histones) or $5 \times 10^7$ cells (ChIP of transcription factors) were cross-linked with 1% formaldehyde for 10 min at room temperature and stored in −80 °C. The nuclei were extracted with Hypotonic Buffer (20 mM HEPES pH 7.5, 10 mM KCl, 1 mM EDTA, 0.2% NP40, 10% Glycerol, 1× Protease Inhibitor Cocktail (PIC; Roche)) and sonicated in Sonication Buffer (20 mM Tris Cl pH 8.0, 2 mM EDTA, 150 mM NaCl, 1% NP40, 0.1% SDS, 1× PIC) using a Misonix Sonicator 3000 (Pulse on: 7 min; Amplitude: 70).

Sheared DNA was incubation with 5–10 µg primary antibody (Santa Cruz: ETS1 SC-350; P300 SC-585; MYC SC-40. Abcam: H3K27ac ab4729; H3K36me3 ab9050; H3K4me1 ab8895; H3K9me3 ab8898. Active Motif: H3K4me3 39159. Bethyl Laboratories: BRD4 A301-985A50. Millipore: H3K27me3 17-622; RNAPII 05-623B) at 4 °C and then incubated with BSA-blocked Dynabeads Protein G for 4 h. Bead-bound DNA was rinsed with three to five times in RIPA Buffer (50 mM HEPES pH 8.0, 1 mM EDTA, 1% NP40, 0.7% sodium deoxycholate, 1% TritonX-100, 1× PIC), de-crosslinked in a 65 °C water bath overnight, digested with RNAase and Proteinase K to remove RNA and protein, respectively, and then purified with the Qiaquick PCR purification kit (Qiagen).

ChiP-seq library preparation was perform as described with NEBNext ChIP-seq Library Prep Master Mix Set for Illumina. Libraries were sequenced (50 bp single end) on an Illumina HiSeq 2500.

**NGS data processing**. FASTQ files of ChiP-seq data were aligned to hg19 using bowtie2[39]. We use MACS 1.4[40] to call ChIP-seq peaks. The read density (reads/nt) around ETS1-induced peaks was calculated using Homer[41] and further normalized by region length and total reads number per library (reads per million). To identify differentially occupied genomic regions, MACS was used with the bdgdiff option.

For RNA-seq, sequenced reads were aligned to hg19 using Tophat2[42], supplemented with ERCC-spike in sequences (http://tools.invitrogen.com/

downloads/ERCC92.fa). The FPKM (fragment per kilobase of transcript sequence per million base pairs sequenced) of each transcript was computed with Cufflinks 2.0[43]. The FPKM of Spike-In sequences was used to renormalize gene expression with loess regression using the loess.normalization function in the R affy package[44].

For calculating signals within promoter regions, we used TSS $\pm$ 1 kb. ETS1 and RNAPII ChIP-seq reads within promoters were normalized by library size, and input read density was subtracted. The ETS1 promoter density was correlated with RNAPII promoter density or gene expression (FPKM). The trend lines were generated using smoothed cubic splines (via smooth.spline in R) with no degrees of freedom. In Fig. 5c, fold change of ETS1 promoter occupancy (ETS1 at 1, 4, 12 h over 0 h) was correlated with fold change of gene expression (FPKM at 1, 4, or 12 h over 0 h) at same time point upon VEGF stimulation. The bin size of ETS1 fold change used for Fig. 5c was 0.2 (log2). The correlation R value was calculated using a linear regression model.

Gene ontology analyses were performed using GREAT[45]. Plots of set intersections were generated using UpSetR.

**PI calculation**. The PI was calculated based on RNAPII ChiP-seq. PI was calculated as described previously[13]. Briefly, the ratio of normalized RNAPII ChiP-seq reads in the TSS region (TSSR; −50bp to + 300 bp around TSS) was divided by the reads in the gene body (+300 bp to 3 kb past the transcription end site): where L1 and L2 are the lengths of the TSS and gene body regions.

The RNAPII and input reads were uniquely aligned to the Refseq genes and normalized against the length of the counting regions and the library size (mapped reads per million bps). For each Refseq region, the normalized RNAPII ChiP-seq reads minus the normalized input reads were used for the final PI calculation. PI was calculated for all genes with transcriptional activity, except where RNAPII ChiP-seq signal was lower than input.

**Data availability**. High throughput data are available on the Cardiovascular Development Consortium server at https://b2b.hci.utah.edu/gnomex/. ChIP-Seq and RNA-Seq data have also been deposited at GEO under accession code GSE93030.

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

## Acknowledgements

We thank Ali Shilatifard (Northwestern University) for kindly providing the ELL2 and AFF4 antibodies, and Jian Ding and Da-Zhi Wang for providing the modRNA cloning vector. B.Z. was funded by the National Science Foundation of China (91539109, 31671503), A Thousand Young Talent Award (16Z127060017), AHA Scientist Development Grant 14SDG20380866, and T32HL007572. D.S.D. was supported by NSF award #1122374. W.T.P. was supported by an AHA Established Investigator Award, NIH 2UM1HL098166, and by charitable donations to the Boston Children's Hospital Department of Cardiology. L.E.H.S. was supported by NIH EY024864, EY017017, and P01HD18655.

## Author contributions

B.Z. conceived of the study. W.T.P., J.C., Y.F., and B.Z. designed the experiments. B.Z., Y.F., S.W., S.M.S., S.W., Y.Z., F.Z., and Y.S. collected the data. J.C., K.S., B.Z., D.S.D., P.J. P., and W.T.P. performed and refined the data analysis. B.Z. and W.T.P. drafted the manuscript and all authors contributed to revising it. All authors read and approved the manuscript.

## Additional information

**Competing interests:** The authors declare no competing financial interests.

