## [Peer Review File · Nature Communications]

Reviewers' comments:

Reviewer #1 (expert in transcriptional regulation of angiogenesis)

Remarks to the Author:

The manuscript by Chen et al focuses on RNA Pol 2 pause-release, a process which is increasingly recognised as important in the regulation of gene expression. The Pu lab has been leading the way in this field, and this excellent study is another novel and important contribution, specifically in the area of endothelial biology and angiogenesis where little has been published so far on this topic. On the whole, it is a very accurate, comprehensive and convincing study. I would recommend toning down the conclusions regarding the functional relevance of this pathway, since in my view the in vivo experiments support but do not conclusively demonstrate that the BRD4 -ETS1 interaction, and hence the RNAPII pause-release pathway, is indispensable in angiogenesis.

Main comments:

- Overall, the main text provides very limited details in terms of experimental design and interpretation, so that the reader has work quite hard. Throughout the manuscript, more details should be provided about the experiments, to make this work more accessible (examples mentioned below, but applicable to most of the results section)
- Because this work is clearly of interest to the endothelial/angiogenesis community, the paper should provide a more detailed introduction to the process of RNA Pol2 pause-release, its importance in development and its basic mechanisms. Equally, the discussion should be expanded on the importance of this process in developed as compared to developing organisms/tissues.
- The introduction should also have a paragraph about ETS1, focusing on the relevant aspects of its biology and crucially its relationship to VEGF signalling in EC.
- Page 4: "Comparison of ETS1 promoter occupancy to gene transcriptional activity showed that ETS1 preferentially occupied promoters of expressed genes, and infrequently occupied promoters of non-transcribed genes (Fig. 1c)" And fig 1c legend: "Heatmap of indicated chromatin features at promoter regions. Regions are ordered by ETS1 binding strength. Features positively correlated with gene expression correlated with ETS1 binding strength." Where does the gene expression data come from, how does the ETS1 global enrichment correlate with the ETS-1 global transcriptome? Is this carried out in resting or VEGF stimulated HUVEC? Please specify and provide details in main text and legend
 - Several studies have shown that constitutive ETS-1 levels and activity in HUVEC are low, and are upregulated by various agents, such as VEGF in this case. The Authors chose to over-express ETS1 in HUVEC. There is a danger in this approach, given the shared "GGAA/T" core motif shared by the many EST factors expressed in EC. Thus over-expression of ETS factors may result in unwanted forced interaction with DNA binding sites for other ETS factors. This is something that could easily be checked, by carrying out ChIP PCR of targets which are regulated by other ETS factors, not ETS1. This is also relevant for the interpretation of fig 3d.
 - Figure 2b measures PI, which is the key to pause release: again, for this readership, a few more details in main text and legend, not only in the methods, would help.
 - Fig 4i-j (and comment to Fig 5c-d): VEGF is known to induce expression of ETS1 – this has been shown by several groups. It is surprising that the Authors do not see this in their study. Can they comment on this puzzling discrepancy?
 - Fig 5a-b: please provide more details in fig legend to understand the data
 - Fig 5c: please provide scale on the right to be able to evaluate the reported increase in ETS1 binding at the KDR locus
 - Fig 5f: I don't find this figure very user-friendly. The main point is to show that ETS1-NT interferes with expression of early and later-response genes. The effect of ETS1-NT is not seen at 1 hour – what does that mean? The acetylation-mimetic mutant has a stronger effect at 1 hr but this is lost at 4 hr – what does this mean? The VEGF timecourse should be taken into account and results described and discussed, alternatively a single time point would suffice.

- Fig 6a -b are uninformative and should be removed. The images could be included in Supplementary possibly to show non-toxicity, but they are not helpful to demonstrate migration. Endothelial migration in transwells is an inadequate and surpassed test for EC motility, since EC are adherent cells. The simplest, accepted in vitro migration assay is the scratch wound assay. Overall, cellular phenotyping using more sophisticated techniques should be used to match the high quality global molecular analysis
- Fig 6d: does the BRD4 inhibitor JQ1 affect cell proliferation or rather induce cell death? The massive reduction in cell numbers would suggest the latter. This would be in line with in vivo data in Fig 7 - and would definitely influence the interpretation of the effect of JQ1 on cell migration (panels a-b).
- Fig 7: please provide evidence of the homogenous endothelial phenotype of ECFC in supplementary, including passage etc
- The Discussion is very brief and a third is dedicated to summarising the findings. It should be expanded, to set these results in the context of what's known about the relevant molecular as well as cellular pathways and also to mention the possible limitations of this study.

Minor comments:

- Despite the many alternative names for ETS that appear on Genecard, the most commonly used is E26 transformation-specific sequence
 - Page 5: "ETS1 modRNA increased ETS1 expression.... without affecting cell morphology (Fig 2a): the figure does not show morphology. They probably mean Supplementary figure?"
 - Page 7, mid page: "to map the interacting BRD4-interacting domain..." – remove the first "interacting"
- I would suggest to move the model from Fig 7d to earlier, since the in vivo data does not add to the model but provides some important functional relevance.

Reviewer #2 (expert in RNAPII pause release)

Remarks to the Author:

The manuscript by Chen et al, entitled, "VEGF promotes RNAPII pause release through ETS1 to stimulate angiogenesis" reports the recruitment of Ets1 to early and late response genes involved in the response of HUVEC cells to VEGF as part of angiogenesis. The authors show that the binding of ETS1 is accompanied by increased Ser2 phosphorylation of RNAPII and a shift of RNAPII into gene bodies decreasing the pausing index. These changes are noted at thousands of genes throughout the genome, and the authors draw a comparison with MYC transcriptome amplification. They show that ETS1 phosphorylation precedes binding of CBP to the transcription factor, that is then acetylated to license BRD4 binding. By mutagenesis (deletions and point mutations, and acetyl-mimetic substitutions for transfection and transfection of Ets1 fragments for competition in vivo and in vitro) the authors show that acetylation is required for BRD4 action. They show that ETS1 activation is sensitive to the BET inhibitor, JQ1.

Overall this is a solid. Well-designed and well-executed and carefully interpreted work. The results are interesting and significant. The manuscript is generally well-written with a few minor exceptions.

The only criticism I have aside from minor textual issues, concerns the somewhat inadequate characterization of the effect on pausing. If the effect on pausing is important enough to be included in the title, then a somewhat fuller characterization of pausing and pause release is to be expected. Some sort of characterization of the distribution of nascent transcripts (for example using Pro-Seq) or a finer resolution mapping of the position of the paused polymerase would seem to be needed. Alternatively, the title should be changed to something that reflects the clear strength of the work, something like "VEGF stimulated ETS-1 acetylation drives transcription enabling angiogenesis".

Also, it would be really nice if the authors could demonstrated acetylation of endogenous ETS1, as opposed to transfected ETS1.

At the bottom of page 10, the authors refer to the BRD4 inhibitor BRD4—I assume they mean JQ1.

The authors need to acknowledge that interpretation of JQ1 inhibition may be complicated by the suppression MYC that is frequently associated with this drug treatment so it may be difficult to deconvolute these effects.

Reviewers' comments:

Reviewer #1 (expertise in transcriptional regulation of angiogenesis)

Remarks to the Author:

The manuscript by Chen et al focuses on RNA Pol 2 pause-release, a process which is increasingly recognised as important in the regulation of gene expression. The Pu lab has been leading the way in this field, and this excellent study is another novel and important contribution, specifically in the area of endothelial biology and angiogenesis where little has been published so far on this topic. On the whole, it is a very accurate, comprehensive and convincing study.

We thank the reviewer for the positive comments.

I would recommend toning down the conclusions regarding the functional relevance of this pathway, since in my view the *in vivo* experiments support but do not conclusively demonstrate that the BRD4 -ETS1 interaction, and hence the RNAPII pause-release pathway, is indispensable in angiogenesis.

We agree that the *in vivo* significance of the BRD4-ETS1 interaction, and more generally RNAPII pause release, requires additional investigation. We more conservatively described the *in vivo* experiments and explicitly acknowledged the need for further study in the revised manuscript: "However, further studies will be required to more comprehensively understand the *in vivo* significance of the VEGF-ETS1-BRD4 signal transduction pathway, or more broadly VEGF regulation of RNAPII pause release, for vessel formation and maintenance."

Main comments:

Overall, the main text provides very limited details in terms of experimental design and interpretation, so that the reader has work quite hard. Throughout the manuscript, more details should be provided about the experiments, to make this work more accessible (examples mentioned below, but applicable to most of the results section) Because this work is clearly of interest to the endothelial/angiogenesis community, the paper should provide a more detailed introduction to the process of RNA Pol2 pause-release, its importance in development and its basic mechanisms. Equally, the discussion should be expanded on the importance of this process in developed as compared to developing organisms/tissues.

In the revised ms, we expanded the introduction to provide more background. We also expanded the discussion to better explain the significance of the findings in the context of the fields of angiogenesis and pausing release. More details were provided in the results and figure legends to facilitate reading the manuscript. The word count limited further exposition.

The introduction should also have a paragraph about ETS1, focusing on the relevant aspects of its biology and crucially its relationship to VEGF signalling in EC.

We have rewrote the introduction and discussion as recommended by the reviewer to introduce ETS1 and RNAPII pausing and their potential function in angiogenesis in greater detail.

Page 4: "Comparison of ETS1 promoter occupancy to gene transcriptional activity showed that ETS1 preferentially occupied promoters of expressed genes, and infrequently occupied promoters of non-transcribed genes (Fig. 1c)" And fig 1c legend: "Heatmap of indicated

chromatin features at promoter regions. Regions are ordered by ETS1 binding strength. Features positively correlated with gene expression correlated with ETS1 binding strength.” Where does the gene expression data come from, how does the ETS1 global enrichment correlate with the ETS-1 global transcriptome? Is this carried out in resting or VEGF stimulated HUVEC? Please specify and provide details in main text and legend

We previously reported RNA-seq data for the same experimental design (HUVEC stimulation for 12 hours with time points at 0, 1, 4, 12 hours){Zhang 2013a}. Fig. 1f shows the positive correlation between ETS1 occupancy and gene expression. All of Fig. 1 refers to the 0 hour time point. These details have been made more clear in the text and figure legend.

Several studies have shown that constitutive ETS-1 levels and activity in HUVEC are low, and are upregulated by various agents, such as VEGF in this case. The Authors chose to over-express ETS1 in HUVEC. There is a danger in this approach, given the shared “GGAA/T” core motif shared by the many EST factors expressed in EC. Thus over-expression of ETS factors may result in unwanted forced interaction with DNA binding sites for other ETS factors. This is something that could easily be checked, by carrying out ChIP PCR of targets which are regulated by other ETS factors, not ETS1. This is also relevant for the interpretation of fig 3d.

This is a good suggestion. We tested this possibility by performed Chip-qPCR of ERG and FLI1, members of ETS family, on GFP or ETS1 modRNA-transfected HUVEC cells. We found that ETS1 overexpression did not change ERG or FLI1 occupancy on most (5 of 7 for each ERG and FLI1) of tested ETS1-upregulated chromatin regions. These data suggest that most changes observed were directly due to increased ETS1 occupancy rather than altered occupancy of other ETS factors. Furthermore, one reason we used ETS1 modRNA for overexpression is that it acutely upregulates ETS1, minimizing the chance for secondary effects on cells. The modRNA experimental endpoints were measured at 12 hours after ETS1 modRNA transfection.

Figure 2b measures PI, which is the key to pause release: again, for this readership, a few more details in main text and legend, not only in the methods, would help.

We added more detail to the main text and figure legend.

Fig 4i-j (and comment to Fig 5c-d): VEGF is known to induce expression of ETS1 – this has been shown by several groups. It is surprising that the Authors do not see this in their study. Can they comment on this puzzling discrepancy?

Watanabe reported that VEGF and hypoxia increased ETS1 mRNA In bovine retinal endothelial cell (BREC){Watanabe 2004}. Similarly we detected increased ETS1 mRNA expression in VEGF treated HUVEC cell by the RNA-seq and RT-qPCR{Zhang 2013a} and in the hypoxia-induced retinopathy model (Supple Fig. 7a). The difference between the two studies is in the protein level. The Watanabe study showed VEGF upregulated ETS1 protein expression in BREC, but the increase (less than 2 fold) was not as strong as observed at the mRNA level. In our study, we did not observe increased ETS1 protein in multiple experiments, despite consistently finding increased ETS1 mRNA. We reason this discrepancy might be attributable to post-transcriptional regulation of ETS1 protein levels. The different dose of VEGF and the specific cell lines studied may also contribute to variation between studies.

Fig 5a-b: please provide more details in fig legend to understand the data

This is an “UpSet” plot, which is a way to visualize the overlaps between four or more sets.

The plot is better explained in the revised figure legend.

Fig 5c: please provide scale on the right to be able to evaluate the reported increase in ETS1 binding at the KDR locus

The revised figure now includes a scale bar.

Fig 5f: I don't find this figure very user-friendly. The main point is to show that ETS1-NT interferes with expression of early and later-response genes. The effect of ETS1-NT is not seen at 1 hour – what does that mean? The acetylation-mimetic mutant has a stronger effect at 1 hr but this is lost at 4 hr – what does this mean? The VEGF timecourse should be taken into account and results described and discussed, alternatively a single time point would suffice.

Figure 5f does present a large amount of data. It investigates the effect of 8 treatments on 8 genes at 4 time points. A heatmap is the best way to show this dataset.

VEGF was for the entire 12 hour time course; samples were withdrawn for measurements at 0, 1, 4, and 12 hours of VEGF exposure. In general the ERG genes and the LRG responded similarly to the 8 treatments. The NT peptide did have an effect at 1 hour (note the heatmap is less red for the ERG genes), NT(KQ) was more potent (heatmap even less red), and NT(KR) was less potent, ie dominant negative activity correlated with BRD4 affinity. These dominant negative peptides also affected the LRG genes with the same potency order.

The reviewer noted a difference in the responses of ERGs to dominant negative peptides at 4 and 12 hours. This led us to review the prior heatmap and we noticed that the scale for upregulation and downregulation was not symmetric. We corrected this in the current heatmap. This change did not affect the major conclusions from these expression data.

Fig 6a -b are uninformative and should be removed. The images could be included in Supplementary possibly to show non-toxicity, but they are not helpful to demonstrate migration. Endothelial migration in transwells is an inadequate and surpassed test for EC motility, since EC are adherent cells. The simplest, accepted in vitro migration assay is the scratch wound assay. Overall, cellular phenotyping using more sophisticated techniques should be used to match the high quality global molecular analysis

As suggested by the reviewer, we further tested HUVEC migration by wound healing assay and observed similar results to the transwell assay of cell migration (Fig. 6C and Suppl. Fig 5e). This further reinforced the indispensable role of ETS1-Brd4 interaction in regulating EC migration. We moved the representative images of these experiments to the Supplement (Suppl Fig. 5c-e)

Fig 6d: does the BRD4 inhibitor JQ1 affect cell proliferation or rather induce cell death? The massive reduction in cell numbers would suggest the latter. This would be in line with in vivo data in Fig 7 - and would definitely influence the interpretation of the effect of JQ1 on cell migration (panels a-b).

We evaluated the toxicity of JQ1 by tunnel assay. The result demonstrated that JQ1 at 500 nM, the dose used in this study, did not show significant pro-apoptotic effects. Above 1000 nM, the apoptotic rate increased (Reviewer Fig.1). Relatively low toxicity of JQ1 was also observed in animal experiments {Filippakopoulos 2010;Anand 2013}. These data suggest that JQ1 at the concentrations used in this study negligibly increases apoptotic cell death. Coupled with our direct measurement of EdU uptake (Fig. 6e), the data indicate that the massive reduction of cell number shown in Fig 6d is cell cycle arrest.

Reviewer Fig.1 TUNEL staining of JQ1-treated HUVEC cells. 500 nM JQ1 did not elevate HUVEC cell apoptotic rate.

Fig 7: please provide evidence of the homogenous endothelial phenotype of ECFC in supplementary, including passage etc.

We used ECFCs around passage 8. We performed immunofluorescent staining and observed that these ECFCs uniformly expressed typical EC markers of KDR, vWF, CDH5 and PECAM1, demonstrating their homogenous EC identity (Suppl. Fig 6a).

The Discussion is very brief and a third is dedicated to summarising the findings. It should be expanded, to set these results in the context of what's known about the relevant molecular as well as cellular pathways and also to mention the possible limitations of this study.

We considerably expanded discussion to better discuss the significance of the findings in the context of work in this field. We also mention limitations of the study.

Minor comments:

Despite the many alternative names for ETS that appear on Genecard, the most commonly used is E26 transformation-specific sequence

We put the full name of ETS in the introduction. We appreciate this suggestion.

Page 5: "ETS1 modRNA increased ETS1 expression.... without affecting cell morphology (Fig 2a): the figure does not show morphology. They probably mean Supplementary figure?"

In Supplementary Figure 2b, the DIC images show transfected cells have typical cobblestone morphology and few death, which provide evidence supporting our claim that ETS1 modRNA transfection did not affect cell morphology significantly.

Page 7, mid page: "to map the interacting BRD4-interacting domain..." – remove the first "interacting"

Corrected.

I would suggest to move the model from Fig 7d to earlier, since the in vivo data does not add to the model but provides some important functional relevance.

The in vivo data provide important functional read-out and evidence for the ETS1-induced pausing release model. Therefore, we prefer to retain the schematic model figure at the end of the paper to better summarize of the entire story. Nevertheless, we appreciated the suggestion.

Reviewer #2 (expertise in RNAPII pause release)

Remarks to the Author:

The manuscript by Chen et al, entitled, “VEGF promotes RNAPII pause release through ETS1 to stimulate angiogenesis” reports the recruitment of Ets1 to early and late response genes involved in the response of HUVEC cells to VEGF as part of angiogenesis. The authors show that the binding of ETS1 is accompanied by increased Ser2 phosphorylation of RNAPII and a shift of RNAPII into gene bodies decreasing the pausing index. These changes are noted at thousands of genes throughout the genome, and the authors draw a comparison with MYC transcriptome amplification. They show that ETS1 phosphorylation precedes binding of CBP to the transcription factor, that is then acetylated to license BRD4 binding. By mutagenesis (deletions and point mutations, and acetyl-mimetic substitutions for transfection and transfection of Ets1 fragments for competition in vivo and in vitro) the authors show that acetylation is required for BRD4 action. They show that ETS1 activation is sensitive to the BET inhibitor, JQ1.

Overall this is a solid. Well-designed and well-executed and carefully interpreted work. The results are interesting and significant. The manuscript is generally well-written with a few minor exceptions.

The only criticism I have aside from minor textual issues, concerns the somewhat inadequate characterization of the effect on pausing. If the effect on pausing is important enough to be included in the title, then a somewhat fuller characterization of pausing and pause release is to be expected. Some sort of characterization of the distribution of nascent transcripts (for example using Pro-Seq) or a finer resolution mapping of the position of the paused polymerase would seem to be needed. Alternatively, the title should be changed to something that reflects the clear strength of the work, something like “VEGF stimulated ETS-1 acetylation drives transcription enabling angiogenesis”.

Utilization of ChiP-seq to characterize RNAPII pausing is widely accepted by the pausing field and much easier to carry out than GRO-seq or PRO-seq. In our previous published paper characterizing pausing state in 35 mammalian cells {Day et al., 2016, #77692}, we evaluate the RNAPII ChiP-seq and GRO-seq (An old version of Pro-seq) and their signals at gene coding regions are well correlated. PI calculated from ChiP-seq is also comparable to GRO-seq. We intensively tried GRO-seq in the lab with a protocol provided by Leighton Core in Jon Lis's lab but were not able to obtain reproducible results. Therefore, considering the technical barrier of perform GRO-seq and evidence that comparable conclusions related RNAPII pause release can be drawn from these techniques, we used RNAPII ChiP-seq to measure the pausing state within ECs in this study.

As suggested by the reviewer, we changed the title to, “VEGF amplifies transcription through ETS1 acetylation to enable angiogenesis.”

Also, it would be really nice if the authors could demonstrated acetylation of endogenous ETS1, as opposed to transfected ETS1.

We measured endogenous ETS1 acetylation in HUVEC by Mass Spectroscopy and also found K8 and 18 acetylation (Suppl Fig 4).

At the bottom of page 10, the authors refer to the BRD4 inhibitor BRD4—I assume they mean JQ1.

We corrected this error.

The authors need to acknowledge that interpretation of JQ1 inhibition may be complicated by the suppression MYC that is frequently associated with this drug treatment so it may be difficult to deconvolute these effects.

JQ1 affects MYC levels because BRD4 stimulates RNAPII pause release of MYC{Delmore 2011}. Therefore the effect of JQ1 on MYC is in keeping with its broader mechanism of action – inhibiting RNAPII pause release by binding BRD4{Filippakopoulos 2010}. Because MYC also stimulates RNAPII pause release, this represents a feed-forward loop and it is difficult to separate the effects of JQ1 on RNAPII pause release from those mediated specifically by MYC depletion.

We acknowledge the complex potential feedback loops and the limitations of the current data to deconvolute them in the revised discussion, “The relationship between RNAPII pause release and MYC, and co-occupancy between MYC and ETS1, suggests possible complex regulatory circuits which are not disentangled by our current study and which will be a fruitful area for further investigation.”

REVIEWERS' COMMENTS:

Reviewer #1 (Remarks to the Author):

The Authors have addressed the Reviewers' points in a satisfactory manner. A few minor comments:

1. The final 6 lines at the end of the Result section, starting with "Moreover", are more suited to the Discussion than to the Results section.
2. The new extended discussion is interesting and relevant; it is however surprising that there is no mention of a paper by Kaikkonen et al (NAR 2014;42:12570), entitled "Control of VEGF-A transcriptional programs by pausing and genomic compartmentalization", which provides a global map of the effect of VEGF-A on paused RNAPol II, and focuses on ETS1 and other TF. The Authors should briefly explain how their work relates to the previous findings by Kaikkonen et al
3. Discussion Page 16, bottom of the page: add "the small molecule inhibitor of BRD4" to "JQ1"

With these minor changes, in my view the manuscript is suitable for publication.

Reviewer #2 (Remarks to the Author):

The authors have noticeably improved the manuscript and this is a valuable contribution. My only issue is that the authors should somewhere note and consider that RNAP-CTD Ser-2 phosphorylation can be conducted by BRD4 that may directly influence pause release and early elongation. So that ETS recruitment of BRD4 may stimulated pause release via several mechanisms.

BRD4 is an atypical kinase that phosphorylates serine2 of the RNA polymerase II carboxy-terminal domain. Devaiah BN, Lewis BA, Cherman N, Hewitt MC, Albrecht BK, Robey PG, Ozato K, Sims RJ 3rd, Singer DS. Proc Natl Acad Sci U S A. 2012 May 1;109(18):6927-32

RNA Polymerase II Regulates Topoisomerase 1 Activity to Favor Efficient Transcription. Baranello L, Wojtowicz D, Cui K, Devaiah BN, Chung HJ, Chan-Salis KY, Guha R, Wilson K, Zhang X, Zhang H, Piotrowski J, Thomas CJ, Singer DS, Pugh BF, Pommier Y, Przytycka TM, Kouzine F, Lewis BA, Zhao K, Levens D. Cell. 2016 Apr 7;165(2):357-71. doi: 10.1016/j.cell.2016.02.036

REVIEWERS' COMMENTS:

Reviewer #1 (Remarks to the Author):

The Authors have addressed the Reviewers' points in a satisfactory manner. A few minor comments:

1. The final 6 lines at the end of the Result section, starting with “Moreover”, are more suited to the Discussion than to the Results section.

We combined these sentences with the last paragraph of the discussion.

2. The new extended discussion is interesting and relevant; it is however surprising that there is no mention of a paper by Kaikkonen et al (NAR 2014;42:12570), entitled “Control of VEGF-A transcriptional programs by pausing and genomic compartmentalization”, which provides a global map of the effect of VEGF-A on paused RNAPol II, and focuses on ETS1 and other TF. The Authors should briefly explain how their work relates to the previous findings by Kaikkonen et al.

We thank the reviewer for pointing out this relevant study, which we now cited in the discussion. In brief, this study showed that VEGF globally influences RNAPII pausing release, and that ETS1 motifs were enriched at regulatory elements that interact with VEGF-regulated promoters. However, the mechanistic links between VEGF, ETS1, and RNAPII pausing release that is central to our current manuscript were not addressed in the Kaikkonen study. Thus, the Kaikkonen study is entirely consistent with our study, which reveals molecular mechanisms that underlie Kaikkonen's observations.

3. Discussion Page 16, bottom of the page: add “the small molecule inhibitor of BRD4” to “JQ1”

We added this phrase as suggested.

With these minor changes, in my view the manuscript is suitable for publication.

Reviewer #2 (Remarks to the Author):

The authors have noticeably improved the manuscript and this is a valuable contribution. My only issue is that the authors should somewhere note and consider that RNAP-CTD Ser-2 phosphorylation can be conducted by BRD4 that may directly influence pause release and early elongation. So that ETS recruitment of BRD4 may stimulate pause release via several mechanisms.

BRD4 is an atypical kinase that phosphorylates serine2 of the RNA polymerase II carboxy-terminal domain. Devaiah BN, Lewis BA, Cherman N, Hewitt MC, Albrecht BK, Robey PG, Ozato K, Sims RJ 3rd, Singer DS. Proc Natl Acad Sci U S A. 2012 May 1;109(18):6927-32

RNA Polymerase II Regulates Topoisomerase 1 Activity to Favor Efficient Transcription.

Baranello L, Wojtowicz D, Cui K, Devaiah BN, Chung HJ, Chan-Salis KY, Guha R, Wilson K, Zhang X, Zhang H, Piotrowski J, Thomas CJ, Singer DS, Pugh BF, Pommier Y, Przytycka TM, Kouzine F, Lewis BA, Zhao K, Levens D. Cell. 2016 Apr 7;165(2):357-71. doi: 10.1016/j.cell.2016.02.036

We thank the reviewer for this suggestion. We referred to this additional mechanism of BRD4 action in the introduction and discussion and referenced these manuscripts.